# A Virtual Geostationary Ocean Color Sensor to Analyze the Coastal Optical Variability

**Marco Bracaglia [1,2,\*], Rosalia Santoleri [1], Gianluca Volpe [1], Simone Colella [1], Mario Benincasa [1] and Vittorio Ernesto Brando [1]**

1    Istituto di Scienze Marine (CNR-ISMAR), Via Fosso del Cavaliere 100, 00133 Rome, Italy;
     rosalia.santoleri@cnr.it (R.S.); gianluca.volpe@cnr.it (G.V.);
     simone.colella@cnr.it (S.C.); mario.benincasa@artov.ismar.cnr.it (M.B.); vittorio.brando@cnr.it (V.E.B.)

2    Dipartimento di Scienze e Tecnologie, Università degli Studi di Napoli Parthenope, Via Amm. F. Acton 38,
     80133, Naples, Italy

\*    Correspondence: marco.bracaglia@artov.ismar.cnr.it

**Abstract:** In the coastal environment the optical properties can vary on temporal scales that are shorter than the near-polar orbiting satellite temporal resolution (~1 image per day), which does not allow capturing most of the coastal optical variability. The objective of this work is to fill the gap between the near-polar orbiting and geostationary sensor temporal resolutions, as the latter sensors provide multiple images of the same basin during the same day. To do that, a Level 3 hyper-temporal analysis-ready Ocean Color (OC) dataset, named Virtual Geostationary Ocean Color Sensor (VGOCS), has been created. This dataset contains the observations acquired over the North Adriatic Sea by the currently functioning near-polar orbiting sensors, allowing approaching the geostationary sensor temporal resolution. The problem in using data from different sensors is that they are characterized by different uncertainty sources that can introduce artifacts between different satellite images. Hence, the sensors have different spatial and spectral resolutions, their calibration procedures can have different accuracies, and their Level 2 data can be retrieved using different processing chains. Such differences were reduced here by adjusting the satellite data with a multi-linear regression algorithm that exploits the Fiducial Reference Measurements data stream of the AERONET-OC water-leaving radiance acquired at the Acqua Alta Oceanographic Tower, located in the Gulf of Venice. This work aims to prove the suitability of VGOCS in analyzing the coastal optical variability, presenting the improvement brought by the adjustment on the quality of the satellite data, the VGOCS spatial and temporal coverage, and the inter-sensor differences. Hence, the adjustment will strongly increase the agreement between the satellite and in situ data and between data from different near-polar orbiting OC imagers; moreover, the adjustment will make available data traditionally masked in the standard processing chains, increasing the VGOCS spatial and temporal coverage, fundamental to analyze the coastal optical variability. Finally, the fulfillment by VGOCS of the three conditions for a hyper-temporal dataset will be demonstrated in this work.

**Keywords:** fiducial reference measurements; hyper-temporal dataset; optical radiometry; coastal environment; observation geometry

## 1. Introduction

Ocean Color Radiometry (OCR) satellite data allow observing the variability of the inherent optical properties (IOPs) and the concentrations of the water components in the coastal and open ocean with a large spatial coverage [1].

Near-polar orbiting OC satellites usually have a spatial resolution that goes from 300 m to 1000 m and provides ~1 image per day of basins located at middle latitudes. These spatial and temporal resolutions are of the same order as the temporal and spatial scales of most of the open ocean bio-optical processes and, consequently, are sufficient to properly characterize such areas [2–4]. On the contrary, in coastal waters, meteorological, marine, and hydrological drivers set up shorter temporal scales for such processes, compared to the open ocean [4]; indeed, in those areas, rapid changes can be observed in the time scale of few hours or less [4–8]. Consequently, an improved temporal resolution than what is currently provided by existing near-polar orbiting sensors is needed to accurately analyze the coastal environment [2–4].

An improved temporal resolution can be provided by OCR geostationary sensors [9–15], which acquire multiple observations of the same area during the same day. Nevertheless, the only currently working OCR geostationary sensor is the Geostationary Ocean Color Imager (GOCI) [16] over the Korean Peninsula, while such sensors are absent over the European Seas.

The first step in improving the near-polar orbiting sensor temporal resolution has been accomplished in [5], where the overlapping scenes of the Visible Infrared Imaging Radiometer Suite (VIIRS), mounted on the SUOMI-NPP satellite, have been exploited to observe the short term variability of some OC quantities in the Gulf of Mexico. This approach has then been used in [6] to observe the particulate backscattering ($b_{bp}$) variability in the North Adriatic Sea (NAS, Figure 1), adjusting the remote sensing reflectance ($R_{rs}$) spectra following [17]. This adjustment, based on a multi-linear regression (MLR) procedure and the AERONET-OC Acqua Alta Oceanographic Tower (AAOT) [18,19] in situ radiometric data (Figure 1), was used to reduce the uncertainties introduced by the different viewing geometry of different VIIRS images and to use data generally masked in the standard processing chains. The AERONET-OC in situ data are part of the Fiducial Reference Measurements (FRM) data stream as they guarantee traceability, accuracy, long-term stability, and cross-site consistency [20]. Those aspects are central for system vicarious calibration, product validation, uncertainty estimation, and development and assessment of bio-optical models for the generation of derived high-level products [21–26].

The present work aims to reduce the gap between near-polar orbiting and geostationary mission temporal resolutions, creating a Level 3 hyper-temporal analysis-ready OC dataset [27,28], containing observations provided by the currently functioning near-polar orbiting OC imagers.

This dataset is defined as an analysis-ready dataset as it is conceived to provide to users, without the expertise in pre-processing procedures, "ready to use" OC data to analyze the coastal optical variability at the high temporal resolution, similarly to the CEOS Analysis Ready Data for Land (CARD4L) [27].

The definition of a hyper-temporal dataset is based on the three conditions provided in [29], for which a hyper temporal dataset must:

(a) Be univariate (i.e., with multiple images of the same parameter only).
(b) Contain a set of time slices, all of which must be precisely coregistered (with image-to-image pixels perfectly aligned spatially).
(c) Exhibit radiometric consistency between images (i.e., they are measured using the same sensors or inter-validated sensor systems, and exhibit a degree of normalization between time slices).

This dataset is named Virtual Geostationary Ocean Color Sensor (VGOCS). It contains multiple images acquired during the same day by the currently functioning near-polar orbiting OC imagers over the NAS study area, allowing approaching the temporal resolution of a geostationary satellite. For each sensor, the time series starts for the first day of acquisition until 31 October 2019.

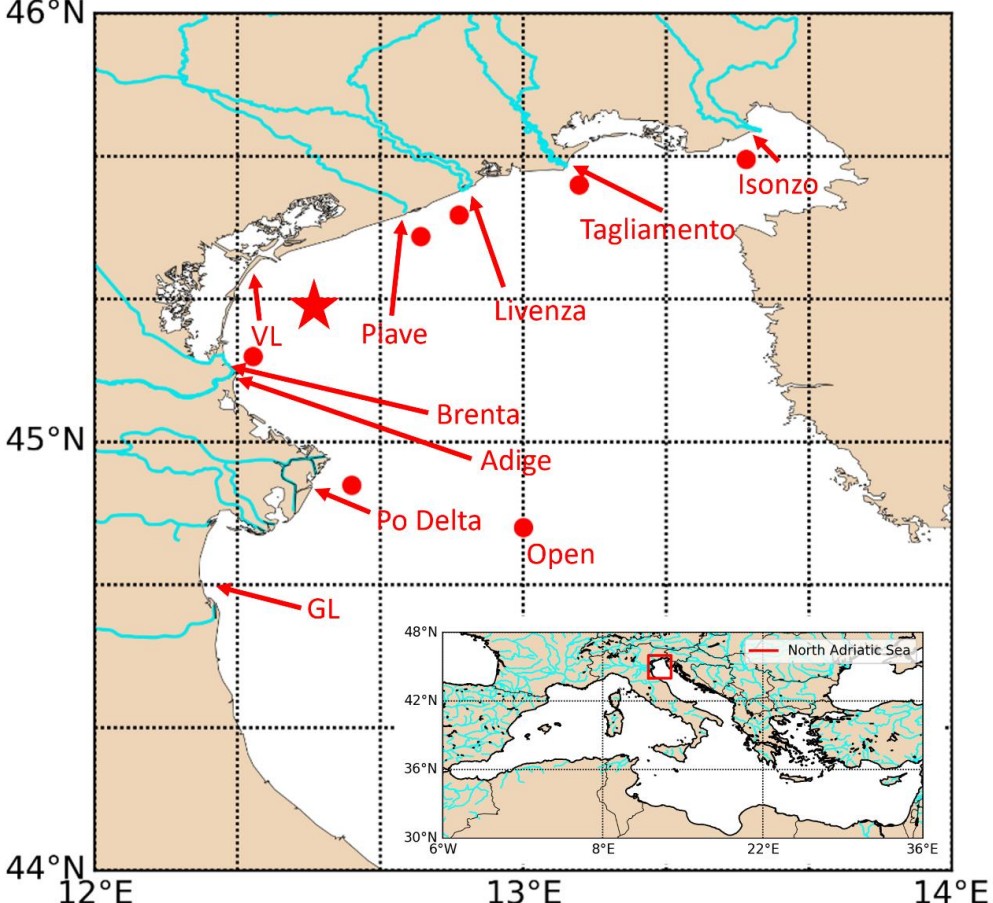

**Figure 1.** Map of the NAS with the rivers of the basin. The red star identifies the Acqua Alta Oceanographic Tower (AAOT), VL the Venice Lagoon, GL the Goro Lagoon, the red dots the virtual buoys (Section 2.7). In the box on the bottom right the Mediterranean Sea, with the NAS identified by the red square.

The creation of this dataset started from the Level 2 (L2) $R_{rs}$ spectra provided by National Aeronautics and Space Administration (NASA) and the European Organisation for the Exploitation of Meteorological Satellites (EUMETSAT); the radiometric data of all sensors have been preserved at their native spectral resolution and re-projected on a common 1 km × 1 km equirectangular grid. The grid spatial resolution of 1 km is probably too coarse for the short spatial scale processes that can take place in the coastal environment [4,30–33]; nevertheless, this is the best that is possible to achieve for the creation of a hyper-temporal dataset created with several OCR sensors. After the re-projection, the $R_{rs}$ spectra, adjusted following [17], have been used to calculate various IOPs by the Quasi Analytical Algorithm version 6 (QAA) [34,35]. This algorithm has been chosen as it has been demonstrated suitable for the $b_{bp}$ retrieval in the NAS [6,36,37].

The adjustment presented in [17] is here exploited to use data from different sensors, as its application to the $R_{rs}$ spectra can reduce the artifacts that can be observed between different satellite images. Indeed, the use of data from different sensors introduces an additional source of uncertainty, besides the one due to the different viewing geometry of each satellite image [6,38]. Hence, the sensors have different spatial and spectral resolutions, their calibration procedures can have different accuracies, and their L2 data can be retrieved using different processing chains. Consequently, looking at two scenes acquired during the same day by different sensors, it is not easy to discern how much of the observed difference is due to a real process or the artifacts, introduced by the different data uncertainties [8,39].

The objective of this work is to prove:

- The suitability of VGOCS in analyzing the coastal optical variability, by the study of the effect of the adjustment on the quality of the satellite data, the VGOCS spatial and temporal coverage, and the intersensor differences.
- The fulfillment of the three conditions for a hyper-temporal dataset [29].

The manuscript is structured as follows: in the next section the study area, the material, and the methods used in this work will be presented, while in Section 3 the obtained results will be analyzed. In the latter section, the results of the match-up analyses between different satellite and in situ data before and after the application of the adjustment will be presented. Then the spatial and temporal coverage and the inter-sensor differences of the adjusted VGOCS data will be analyzed. In the last two sections, the results will be discussed and the conclusions and the future perspective for this work will be presented.

## 2. Materials and Methods

### 2.1. Study Area

The NAS is characterized by a large number of rivers (listed in Figure 1) that can discharge waters with different biogeochemical and optical properties [40–42]. This basin is located in the North-Eastern Mediterranean (Figure 1) and its main rivers are shown in Figure 1. All those rivers are relevant for the optical dynamics of the basin [40,43,44], but Po and Adige are those that provide most of the NAS freshwater [45,46].

The NAS optical variability is strongly influenced by meteo-marine and hydrological forcings, and by the basin shallow bathymetry, which has an average depth smaller than 35 m [41,43–50]. Due to this complexity, several works studied the bio-optical properties and the dynamics of the NAS using several kinds of data and methods such as in situ, ship-board, and satellite observations and oceanographic models [51–56].

### 2.2. VGOCS Sensors

VGOCS is composed of the observations acquired over the NAS by the current functioning near-polar orbiting OC sensors. Those involved in the dataset are

- The NASA Moderate-Resolution Imaging Spectroradiometer (MODIS) sensors mounted on the AQUA and TERRA satellites, here simply referred to as AQUA and TERRA respectively.
- The National Oceanic and Atmospheric Administration (NOAA) VIIRS sensors mounted on the SUOMI-NPP and NOOA-20 (previously JPSS-1) satellites, here referred to as VIIRSN and VIIRSJ respectively.
- The European Space Agency (ESA)/EUMETSAT Ocean and Land Color Instrument (OLCI) mounted on Sentinel 3A, here referred simply as OLCI.

The two MODIS sensors, mounted on the AQUA and TERRA satellites, are the only sensors of the dataset that have the same technical characteristic and resolutions; indeed, the two VIIRS sensors are very similar to each other but differ for the spectral resolution [57]. The latter is depicted for each sensor in Figure 2 [57–60].

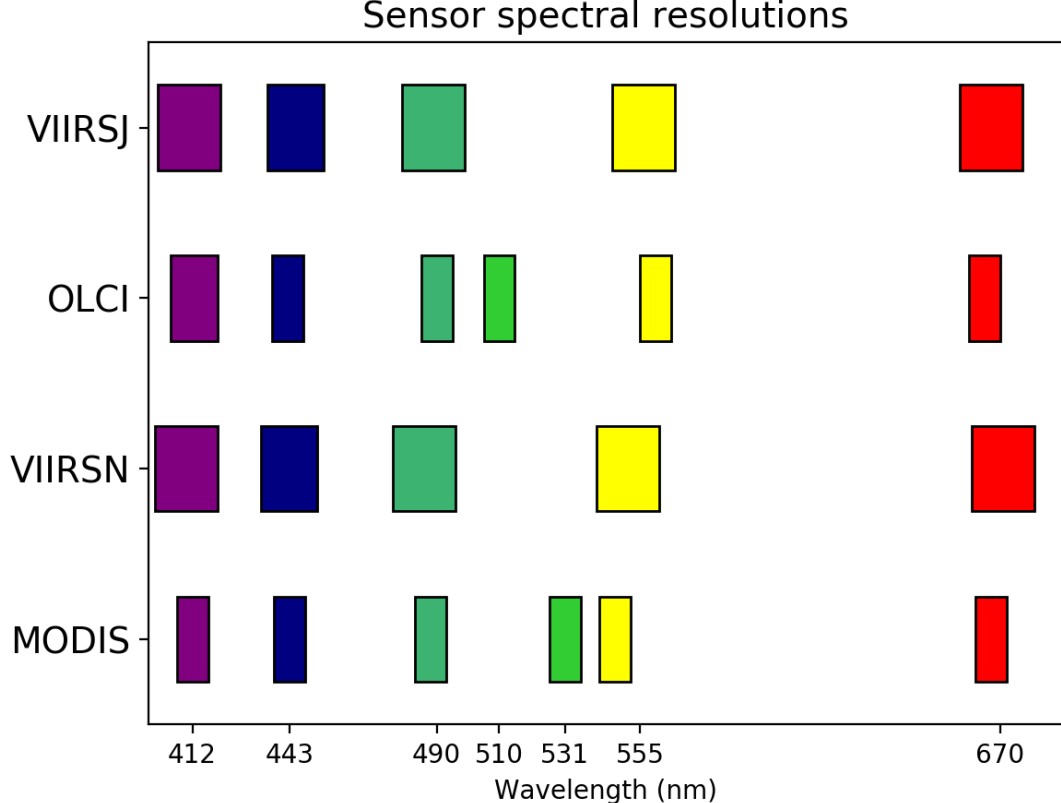

**Figure 2.** Spectral resolutions of the sensor involved in the Virtual Geostationary Ocean Color Sensor (VGOCS).

The MODIS OC bands have a 1000 m spatial resolution [60], the two VIIRS sensors a 750 m one [58], while the OLCI data are provided at two different resolutions: Full resolution (FR, 300 m) and Reduce Resolution (RR, 1200 m) [59]. In this study, the RR products are exploited, as the FR data suffer from a salt and pepper effect that is due to atmospheric correction problems (i.e., a lot of valid observations are surrounded by masked pixels, not shown).

OLCI, which is the only ESA/EUMETSAT sensor used in VGCOS, is the one with the lower revisiting frequency. This sensor has an orbital cycle of 27 days [59], while for the NASA/NOAA (N/N) sensors this is of 16 days [58,60].

OLCI has also a different viewing geometry and engineering than the N/N sensors. Indeed, the swath of the MODIS and VIIRS sensors is centered on the nadir and a rotating mirror reflects the radiation onto a set of Charge-Coupled Device (CCD) detectors. In OLCI, 5 CCD detectors are mounted in a fan arrangement on a common optical bench and the radiance arrives at them after passing a calibration assembly [59]. The sensor has a total field of view of 68.5°, while each detector has a field of view 14.1° with a slight overlap of 0.6° between each other [59]. Moreover, to reduce the sun-glint effect, which impacted a large number of the MEdium Resolution Imaging Spectrometer (MERIS) observations at sub-tropical latitudes [59,61], the OLCI swath is not centered at nadir but is tilted 12.6° westwards [59].

### 2.3. VGOCS Dataset and Processing Chain

The VGOCS dataset is created starting from the standard L2 $R_{rs}$ spectra provided by EUTMESAT (OLCI, processing baseline v2.23) [62–64] and NASA (N/N sensors, processing version R2018.0) [65], that are stored in the archive of CNR ISMAR in Rome [39].

The VGOCS processing chain (Figure 3) is mostly the same as the one used in [6], which was based on the standard Copernicus Marine Environment Monitoring System (CMEMS) operational

processing chain for the Mediterranean Sea [39], with some changes. The main difference is that CMEMS provides a single daily image, derived from the merging of data from different sensors [39], while in the VGOCS processing chain the merging procedure is not performed. Hence, all the images are retained to have multiple satellite observations of the basin during the same day.

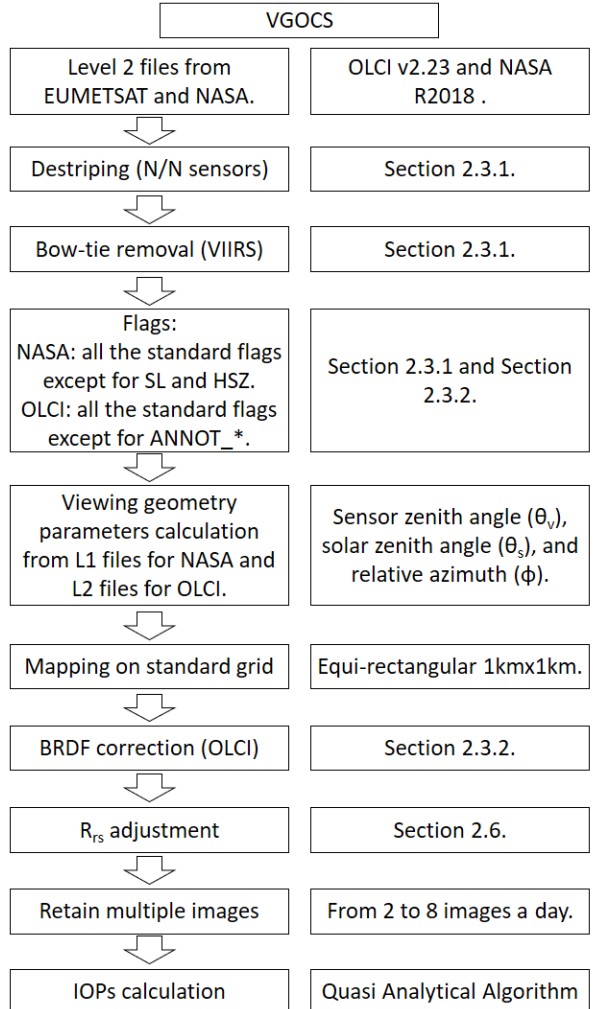

**Figure 3.** On the left the flow chart of the VGOCS processing chain, on the right some additional information for each step.

VGOCS contains observations of the NAS from 2002 (AQUA and TERRA operating) to 31 October 2019 (all sensors operating), providing 2 to 8 images a day depending on the number of functioning sensors and on the day of the satellite orbital cycles. The latter parameter is useful as for each sensor the number of images and their viewing geometry is the same for the same orbital cycle day, except for some minor changes in the ground-track and overpass times. The orbital cycle day is calculated in the same way for all the N/N sensors and it is a number that goes from 1 to 16. Hence, all the N/N sensors involved in VGOCS have an orbital cycle of 16 days [58,60]. For each file, defining as NDT the number of days between the satellite observation and the first TERRA acquisition (8 February 2000), the orbital cycle day can be defined as the remainder of NDT/16 plus one. The orbital cycle of OLCI is of 27 days, thus the orbital cycle day is defined as the remainder of NDO/27 plus one (where NDO is the number of days between the satellite observation and the first OLCI acquisition, 26 April 2016).

To use the same grid for each image of the dataset, all of them are re-projected on a standard 1 km × 1 km equirectangular grid by the nearest neighbor approach.

The name format of each file is VGOCS_YYYYJJJHHMMSS_X.nc, with Y, J, H, M, S (year, Julian day of the year, hour, minute, second) referring to the start time of the sensor acquisition, while X is the label which identifies the sensor (J: VIIRSJ, V: VIIRSN, A: AQUA, T: TERRA, O: OLCI).

The parameters that can be found in each VGOCS file are stored in different groups:

- The $R_{rs}$_data group, where the $R_{rs}$ spectra at the native sensor resolution are stored.
- The IOP_data group, where all the parameters (listed below) calculated with the QAA can be found [34,35]:

  ○　　reference wavelength ($\lambda_0$)
  ○　　total absorption at the reference wavelength ($a(\lambda_0)$)
  ○　　particulate backscattering at the reference wavelength ($b_{bp}(\lambda_0)$)
  ○　　$b_{bp}$ spectral slope ($\eta$)
  ○　　$b_{bp}$ at 443 nm ($b_{bp}(443)$)
  ○　　absorption by nonalgal particles and dissolved matter at 443 nm ($a_{dg}(443)$)
  ○　　$a_{dg}$ spectral slope (s)
  ○　　absorption by phytoplankton at 443 nm ($a_{phy}(443)$)

- The Atmospheric_data group contains some atmospheric parameters, such as aerosol optical thickness and angstrom coefficients, extracted from the L2 files.
- The Geo_data group contains information about the applied flags, extracted from the L2 files, and about some viewing geometry parameters, such as the sensor zenith angle ($\theta_v$), the solar zenith angle ($\theta_s$), and the relative angle between the solar and sensor azimuth angle ($\varphi$). Those parameters for the OLCI sensor are extracted from the L2 file, for the VIIRS sensors from the L1 GEO files, while for the MODIS sensors they are retrieved using the L1B files as the input of l2gen.

Moreover, latitude, longitude, and the satellite orbital cycle day are provided as stand-alone variables.

### 2.3.1. NASA/NOAA Sensors

The processing chain used for the N/N sensors is the same reported in [6] for the VIIRSN data. All the N/N sensor images have been destriped following [66,67], while, for the two VIIRS sensors, the bow-tie missing data have been filled by linear interpolation [58]. Not all the standard L2 flags [68] have been applied; hence, the data have been unmasked for the High Sensor Zenith (HISATZEN, here HSZ) and the stray-light flag (STRAYLIGHT, here SL). The first one masks data acquired at a $\theta_v > 60°$ [69,70], while the second one masks data that could be affected by the adjacency effect of clouds and lands [71–74].

The HSZ flag has not been applied because a large number of images are acquired at $\theta_v > 60°$, and unmasking for this flag makes available a larger number of valid pixels. Using the HSZ data is fundamental for the aim for which VGOCS is conceived, as, to analyze the daily optical variability, it is important to have at disposal as many full images as possible. In this study, an image will be considered as "full" if the sensors detected at least 90% of the water pixels of the NAS, while it will be considered as "partial" if the sensor detected between 30% and 90% of the water pixels.

In Figure 4 the number of images for each day of the orbital cycle is presented for the N/N sensors considering as masked the HSZ observations. As can be seen, only for days 5, 8, 13, and 16 there are five full images (in green), while for the other days there are maximum one partial (in black) and four full images.

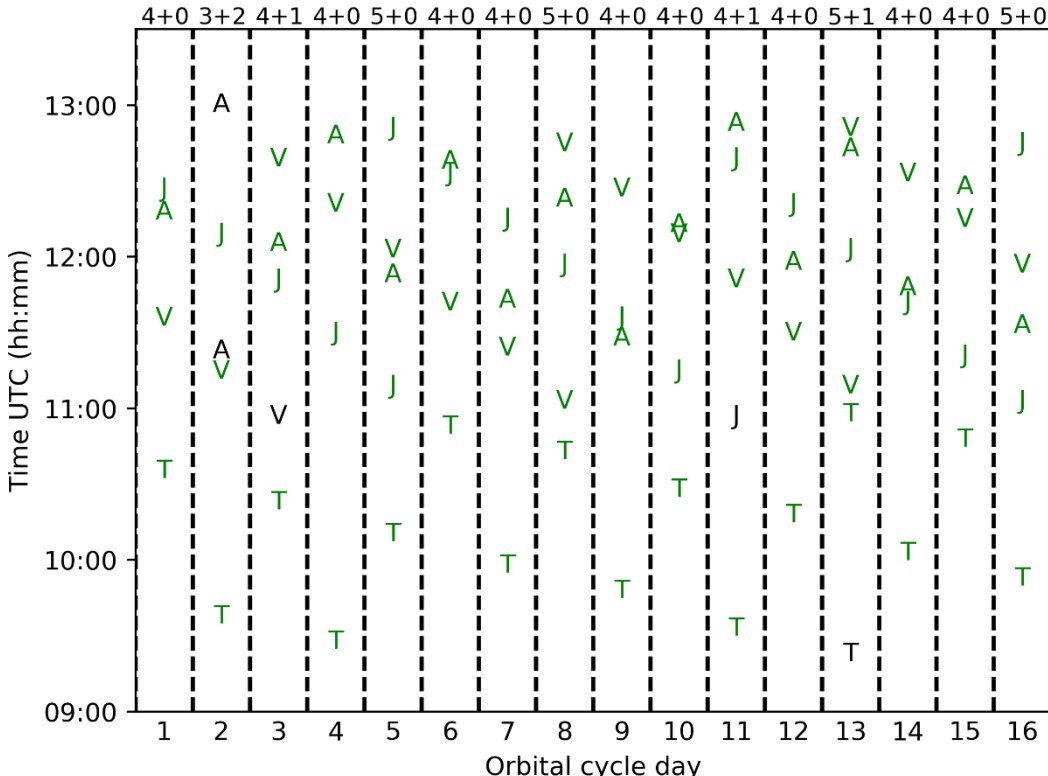

**Figure 4.** Available images for the N/N sensors for each day of the orbital cycle, with the application of the High Sensor Zenith (HSZ) flag. On the x-axes the orbital cycle day, on the y-axes the scan start time, with the different letters identifying the different sensors. In green the full images, in black the partial images. On the top for each orbital cycle day the number of full images plus the number of the partial ones.

Figure 5 is the same of Figure 4, but with the pixels unmasked for the HSZ flag. The HSZ pixels are 22.9% of the pixels scanned by the N/N sensors in the NAS, and unmasking them provides a larger number of full and partial images, in comparison with Figure 4. Now the minimum number of available images is reached on day 3 with 5 full images and the maximum on day 4 with 7 full and 1 partial image. Particularly:

- For VIIRSN there are 9 full orbit overlaps in comparison with the two available maskings for HSZ.
- For VIIRSJ there are 8 full and 2 partial overlaps in comparison with the two available maskings for HSZ.
- For AQUA and TERRA, there are 3 full orbit overlaps (previously zero), and some additional partial images.

Thus, the unmasking for this flag strongly improves the spatial and temporal coverage of VGOCS, making available up to 3 additional full images a day.

It is important to remind here that OLCI has not been taken into account in Figures 4 and 5 as it has a different orbital cycle and no HSZ flag. This sensor can provide one additional complete image of the NAS per day except for days 7, 11, 15, 19, 23, and 27 of its orbital cycle when the NAS is not overpassed by Sentinel-3A.

The SL flag masks all the pixels that could be affected by the light reflected by the land surface that can be forward scattered into the sensor field of view [71–74]. Nevertheless, the application of this flag results in the masking of a large part of the coastal area (here identified in red in Figure 6) [6,75], as can be noted in Figure 7. Here two $b_{bp}(443)$ maps retrieved from the original VIIRSN $R_{rs}$ during 21 March 2013 12:48 UTC are presented: on the left the SL flag is active, and a large portion of the coastal

area results masked; on the right, the SL flag is not applied and the majority of the coastal pixels are now available.

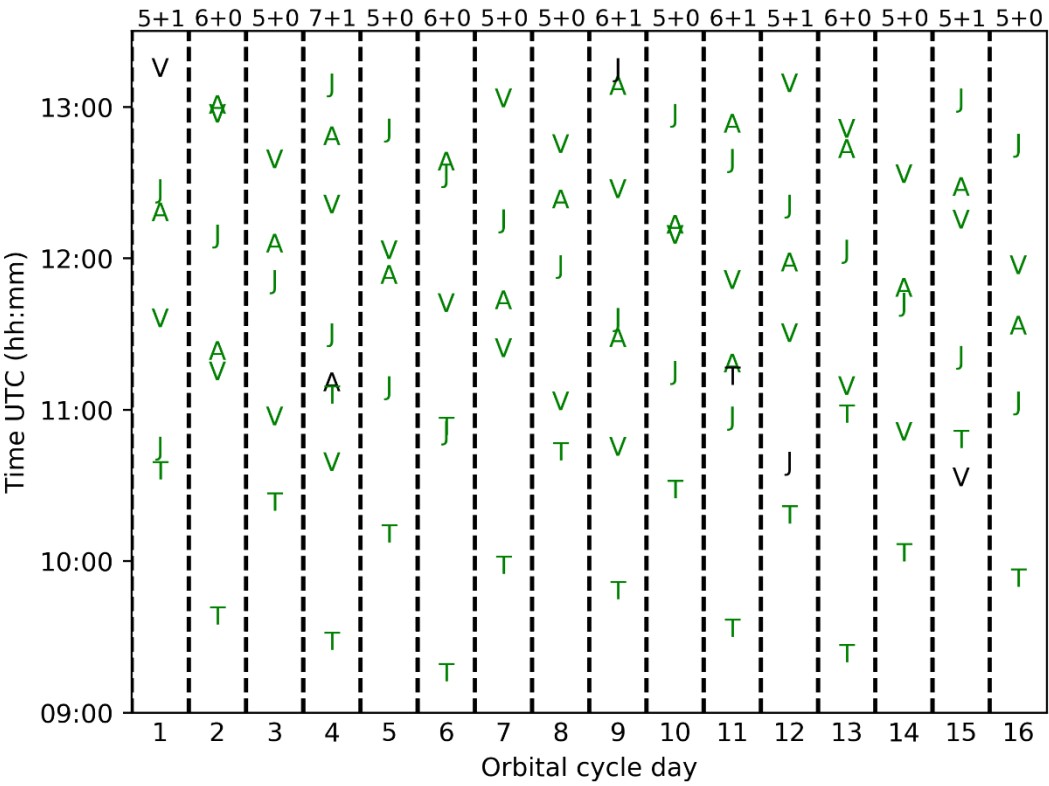

**Figure 5.** Available images for the N/N sensors for each day of the orbital cycle, without the application of the HSZ flag. On the x-axes the orbital cycle day, on the y-axes the scan start time, with the different letters identifying the different sensors. In green the full images, in black the partial images. On the top for each orbital cycle day the number of full images plus the number of the partial ones.

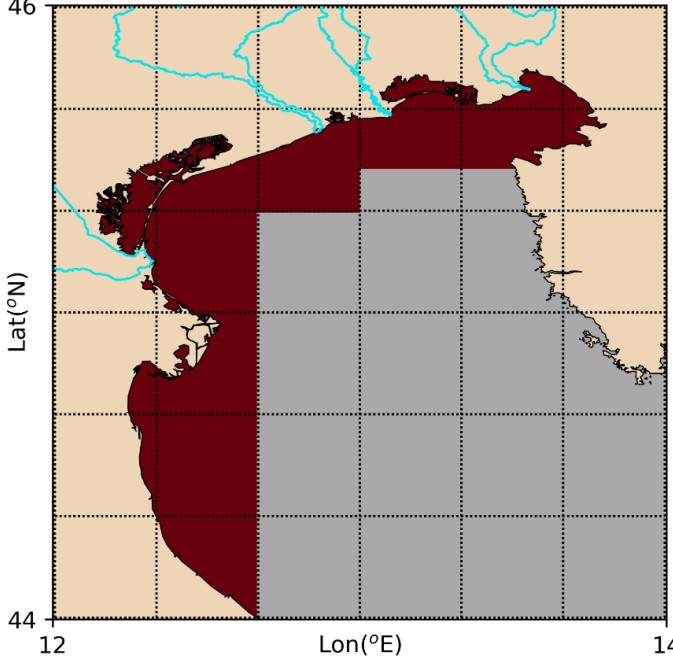

**Figure 6.** Map of the North Adriatic Sea (NAS), in red the coastal area used to quantify the stray-light flag (STRAYLIGHT, here SL) and ANNOT_* effect on the VGOCS spatial coverage.

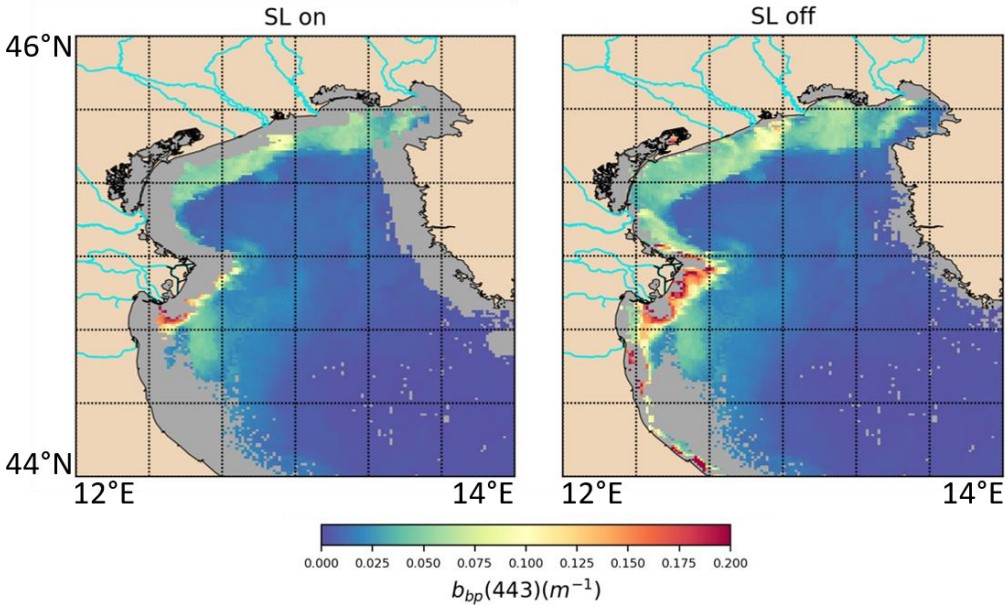

**Figure 7.** Maps of $b_{bp}(443)$ retrieved from the original VIIRSN $R_{rs}$ for 21 March 2013 12:48 UTC. On the left, the SL flag is active (SL on), while the pixels are unmasked (SL off) on the right.

The behavior observed in Figure 7 is common to a large number of images of the dataset, as the mean percentage of the valid pixels flagged by SL in the coastal area is >40% for every single N/N sensor and the entire VGCOS N/N time series (Table 1). Hence, to analyze the coastal optical variability it is necessary to unmask such pixels.

**Table 1.** The mean percentage of SL flagged pixels in comparison with the total number of coastal valid pixels, for each N/N sensor and for the entire N/N VGOCS time series.

| Sensor | Percentage of SL Masked Pixels |
|---|---|
| TERRA | 42.0% |
| AQUA | 41.2% |
| VIIRSN | 45.1% |
| VIIRSJ | 44.7% |
| VGOCS (no OLCI) | 42.4% |

The larger uncertainties generally observed for the HSZ and SL data can be reduced by the adjustment procedure [6]; consequently, the agreement of such data with the in situ observations will be analyzed in Section 3.2.

2.3.2. OLCI Sensor

The EUMETSAT OLCI data have different L2 processing, leading to files with structures and variables differing than those for the N/N sensors. Particularly, a different atmospheric correction algorithm is applied to the TOA radiances; hence, for OLCI, instead of the algorithm from [76], used in the L2 NASA processing, the Baseline Atmospheric Correction (BAC) algorithm [77,78] is exploited. Moreover, the $R_{rs}$ provided in the OLCI L2 files are not corrected for the bi-directional effect. For this reason, for the OLCI data, an additional step is needed in the processing chain (Figure 3) to be coherent with the N/N data. Hence, the OLCI $R_{rs}$ spectra are corrected with the bi-directional reflectance distribution function (BRDF), following the procedure outlined in the works of Morel and his co-workers from 1991 [69,78–81].

The EUMETSAT OLCI L2 flags are different than those used in the NASA L2 processing. In this work, in agreement with [82] and the latest EUMETSAT indication (I. Cazzaniga, pers. comm.) [83],

the ANNOT_DROUT flag is not applied, as in the latest version of CMEMS [84]. This flag masks pixel where the value of the residual for the surface reflectance at 510 nm is above a certain threshold, defined in the climatology. Moreover, in this study, the ANNOTMIXR1, ANNOT_TAU6, and ANNOT_ABSO_D named annotation flags (here ANNOT_*), have not been applied in the match-up analysis to analyze their effect on the VGOCS spatial coverage and their agreement with in situ data. Hence, all those flags concern the quality of the atmospheric correction in open waters, and the application to optically complex waters could be questionable [82,83].

In Figure 8, the maps of $b_{bp}(443)$ retrieved from the original OLCI $R_{rs}$ during 21 September 2017, with and without the application of the ANNOT_DROUT and ANNOT_* flags, are presented. The mean percentage of valid pixels masked by the ANNOT_DROUT flag in the coastal area (Figure 6) is around 40.9% for the OLCI time series, the ANNOT_MIXR1 flag masks the 20.1% of them, while the other two ANNOT_* flags mask the 0.8% of the coastal pixels.

Also for the ANNOT_DROUT and ANNOT_* data, the effect of the adjustment on their agreement with in situ observations will be analyzed in Section 3.2.3 to justify the decision for unmasking data traditionally masked in the standard processing chains.

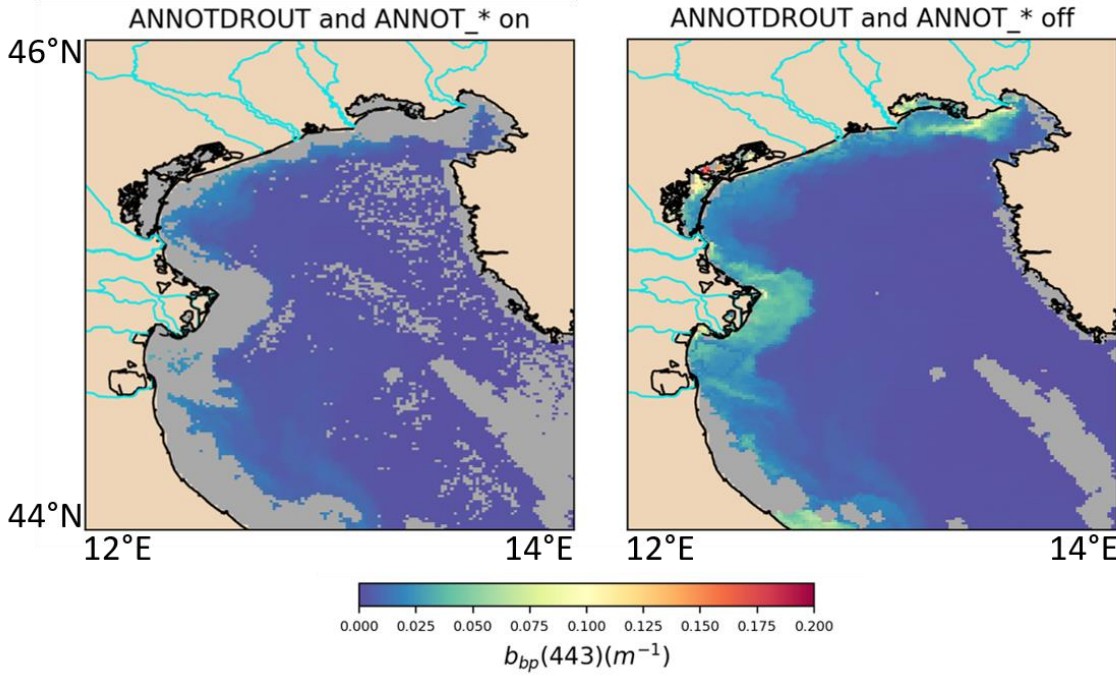

**Figure 8.** Maps of $b_{bp}(443)$ retrieved from the original OLCI $R_{rs}$ for 21 September 2017 09:40 UTC. On the left, the ANNOT_ * flags are active (ANNOT_* on), while the pixels are unmasked on the right (ANNOT_* off).

## 2.4. In Situ Data

AAOT, part of the AERONET-OC network [18,19], is located in the NAS, 8 nautical miles south-east of the Venice Lagoon (12.509° E, 45.314° N, the red star in Figure 1). The in situ data here acquired has been used to validate and adjust the satellite $R_{rs}$ spectra of each sensor.

AERONET-OC can be considered as the archetype FRM for the validation of satellite remote-sensing radiometric data over water [20,85]. Hence, to compare the AAOT data with the satellite $R_{rs}$ spectra, the normalized water-leaving radiances here acquired have been divided by the extra-solar irradiance (F0) [86]. Particularly, in this study the Version 3 L2 AAOT data (cloud-screened and quality assured data) have been exploited and integrated with the L1.5 data (only cloud-screened) for periods where the L2 data were not still processed.

The exploitation of these data to adjust the satellite spectra is fundamental for this study; indeed, AAOT is located in an area with a large optical variability that can be considered as representative of the basin variability [19,87]. Consequently, the adjustment coefficients here retrieved can be used to adjust the $R_{rs}$ spectra in the entire basin [6].

### 2.5. Match-up Analyses

To assess the quality of the satellite $R_{rs}$ spectra, pairwise match-up analyses were performed using the AAOT in situ data as a reference. Those analyses were conducted for every sensor of VGOCS separately, because different sensors can have different sources of uncertainties and, consequently a different agreement with the in situ spectra.

The match-up procedure is consistent with the one of [6], where for each couple of satellite/in situ observations acquired at $\Delta t < 30$ min, a $3 \times 3$ box around the AAOT area has been chosen, keeping only boxes with at least 3 valid pixels. The value of the in situ $R_{rs}$ is the mean of all the AAOT data acquired between t (time of the satellite observation)-$\Delta t$ (30 min) and t+$\Delta t$, while the value of the satellite $R_{rs}$ is the mean of the 3x3 box data For each valid couple of observations, the in situ $R_{rs}$ spectrum has been band-shifted to the satellite wavelengths, following [88].

To test the agreement between the satellite and in situ data, different statistical parameters have been calculated and their formulas have been presented in Table 2.

**Table 2.** Formulations of the statistical parameters used in this study to assess the agreement between two datasets, where n is the number of concurrent observations of the match-up, x the in situ measurement, and y the satellite observation.

| Statistical Parameter | Formulation |
|---|---|
| Determination coefficient ($r^2$) | $r^2 = \dfrac{\sum_{i=1}^{n}(x_i-\bar{x})(y_i-\bar{y})}{\sqrt{\sum_{i=1}^{n}(x_i-\bar{x})^2}\sqrt{\sum_{i=1}^{n}(y_i-\bar{y})^2}}$ |
| Mean Absolute Difference (MAD) | $MAD = \dfrac{\sum_{i=1}^{n}|y_i-x_i|}{n}$ |
| Root Mean Squared Difference (RMSD) | $RMSD = \sqrt{\dfrac{\sum_{i=1}^{n}(y_i-x_i)^2}{n}}$ |
| Mean absolute percentage difference (MAPD) | $MAPD(\%) = \dfrac{100}{n}\sum_{i=1}^{n}\left|\dfrac{y_i-x_i}{x_i}\right|$ |

In the $R_{rs}$ match-up analyses, the entire dataset (ED) of each sensor was divided into two subsets: training (TD) and validation dataset (VD). The first was used to retrieve the coefficients of the adjustment (presented in Section 2.6), while the second one was used to validate the satellite data before and after the adjustment procedure (Section 3). For each N/N sensor (except for VIIRSJ), VD is composed of all the data acquired between 1 September 2015 and 31 October 2019, while the remaining data are part of TD. The split of the dataset is not applied to the OLCI and VIIRSJ data, as they are the youngest sensors between those used in VGOCS and they have a lower number of match up points in comparison with the other sensors. Hence, the splitting may not allow capturing all the optical variability of the basin; consequently, for those two sensors, the data used to retrieve the adjustment coefficients are the same used to validate the dataset (TD=VD=ED).

Match-up analyses have been performed also for data traditionally masked in the standard processing chains. In Section 3.2 match-up analyses for the HSZ, SL, ANNOT_DROUT, and ANNOT_* data have been performed. For the HSZ flag, observation is considered as flagged if the pixel collocated with AAOT is scanned at $\theta v > 60°$; for SL, ANNOT_DROUT, and ANNOT_*, an observation is identified as flagged if the considered flag is active for at least 1/3 of the box valid pixels.

### 2.6. Satellite $R_{rs}$ Adjustment

To reduce discrepancies and biases between in situ and satellite measurements, to exploit data usually masked in the standard processing chains, and to reduce the inter-sensor differences,

a multi-linear regression algorithm (MLR), like the one presented in [17], is exploited to adjust the satellite $R_{rs}$ spectra. As different sensors have different uncertainties, temporal coverages, and agreements with the in situ observations, a different set of adjustment coefficient is calculated for each of them.

Particularly, for each couple of observations of the TD the difference between the in situ and satellite measurement is evaluated for each band:

$$\Delta R_{rs}(\lambda) = R_{rs}^{is}(\lambda) - R_{rs}^{or}(\lambda), \tag{1}$$

where $R_{rs}^{is}(\lambda)$ is the in situ $R_{rs}$ and $R_{rs}^{or}(\lambda)$ is the original satellite $R_{rs}$. The multilinear regression scheme is

$$\langle \Delta R_{rs}(\lambda) \rangle = a_0^{sat} + \sum_{i=1}^{n} a_i^{sat} R_{rs}^{or}(\lambda_i) + a_6^{sat}\theta_v + a_7^{sat}\theta_s + a_8^{sat}\varphi, \tag{2}$$

The input vectors of the adjustment are the original $R_{rs}$ spectra, and the values of the three viewing geometry parameters ($\theta_v$, $\theta_s$, and $\varphi$) corresponding to the pixel collocated with AAOT.

The coefficients $a_i^{sat}$ (i = 0, ... , 8) were calculated performing the MLR scheme between $\Delta R_{rs}(\lambda)$ and the input vectors. Then the adjusted $R_{rs}$ can be retrieved with:

$$R_{rs}^{adj}(\lambda) = R_{rs}^{or}(\lambda) + \langle \Delta R_{rs}(\lambda) \rangle, \tag{3}$$

This set of fixed coefficients have been used to adjust the $R_{rs}$ spectra of each image of the single sensor dataset, as the AAOT in situ data are representative of the optical variability of the basin. Finally, the adjusted $R_{rs}$ have been exploited as the input of the QAA to calculate different IOPs [34,35].

## 2.7. Inter-Sensor Differences

As stated in the previous sections, the adjustment is applied to reduce the $R_{rs}$ inter-sensor differences that are due to different observation geometries, calibration accuracies, data processing, and resolutions of the sensors. To quantify such effect $r^2$ and MAPD have been calculated in different locations of the NAS for images acquired in temporal proximity by different sensors ($\Delta t < 10$ min). This has been done starting from the hypothesis that in a time range of 10 min no considerable oceanographic processes can occur and that consequently most of the differences observed between two images are likely due to artifacts. The chosen locations are here named virtual buoys (red dots in Figure 1) and they are situated:

- At AAOT, to analyze the effect of the adjustment in an area with large optical variability.
- Close to the Livenza, Brenta-Adige, Piave, Tagliamento, and Isonzo river mouths to analyze the effect of the adjustment in optically complex waters.
- In an off-shore location (here named simply OPEN), to analyze the effect of the adjustment in open waters.
- Close to the Po river mouth to analyze the effect of the adjustment in the area where the largest amount of fresh-water is usually available [45,46], due to the high river discharges.

Hence, the last two virtual buoys represent the two optical extreme situations for open and optically complex waters respectively.

As stated earlier, to calculate $r^2$ and MAPD only couples of images acquired at $\Delta t < 10$ min have been selected and for each virtual buoy a $3 \times 3$ box around its location has been chosen. Then only the couples of images with both boxes with at least 3 valid $R_{rs}$ pixels have been maintained, and $r^2$ and MAPD have been calculated for each band of the $R_{rs}$ spectrum, similar to a match-up analysis. As the sensors have a different spectral resolution, in this analysis all the $R_{rs}$ spectra have been band-shifted to the MODIS bands, using the approach from [88].

## 3. Results

### 3.1. Match-Up Analyses

In this section, the agreement between the satellite and in situ $R_{rs}$ spectra before and after the application of the adjustment will be evaluated. In Table 3 the numbers of match-up points for each VGOCS sensor are shown.

**Table 3.** The number of match-up points for each VGOCS sensor. Between parenthesis, for the N/N sensors the number of match-up points for the ED SL affected data, for OLCI those flagged by the ANNOT_DROUT/ANNOT_* flags.

| Sensor | Number of Match-Up Points |
|---|---|
| TERRA | 218 (178) |
| AQUA | 357 (230) |
| VIIRSN | 449 (200) |
| VIIRSJ | 216 (48) |
| OLCI | 64 (37/20) |

Previous studies demonstrated how the MLR adjustment strongly increases the agreement between the in situ and satellite spectra [6,17]. In this work the effect of the adjustment on all the VGOCS sensors will be evaluated but, for the safe of brevity, the scatterplots will be presented only for OLCI. For the N/N sensors, only the spectra of $r^2$ and MAPD before and after the application of adjustment are shown (Figures 9 and 10) and the other statistical parameters (RMSD and MAD) are presented from Tables 4–7).

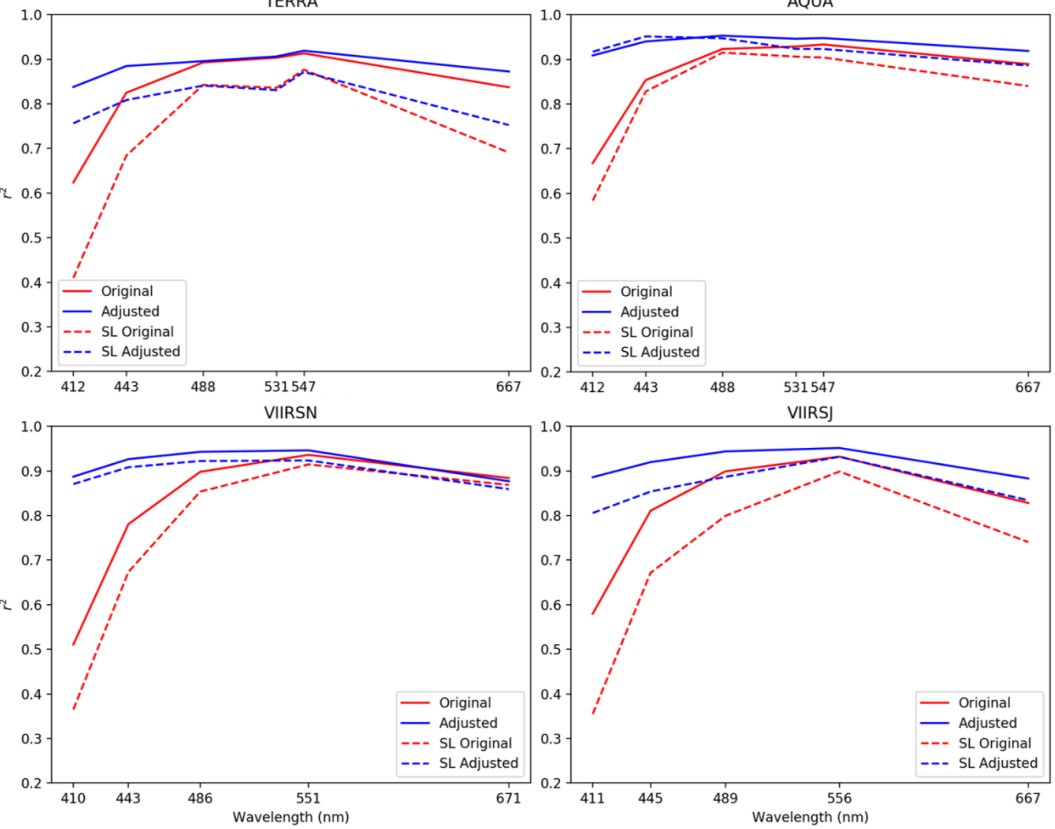

**Figure 9.** Spectra of $r^2$ before (red) and after (blue) the adjustment procedure for each N/N sensor. The continuous lines identified the results for VD, the dotted ones for the ED SL affected data.

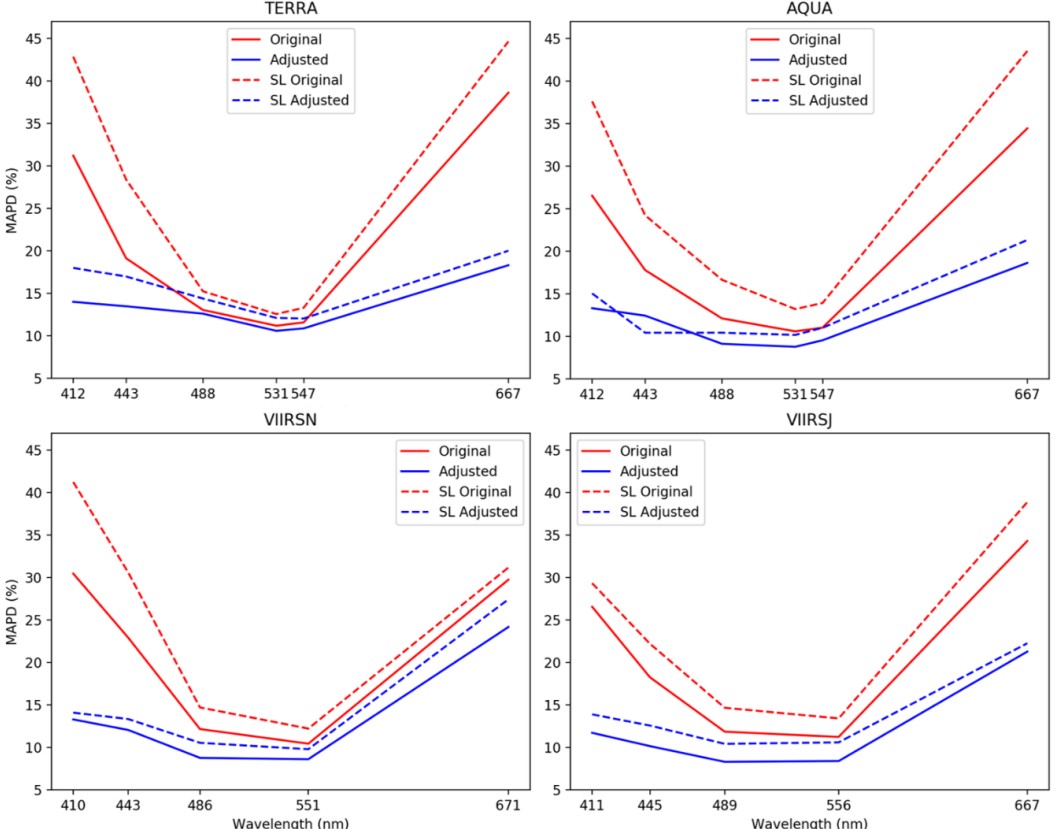

**Figure 10.** Spectra of MAPD before (red) and after (blue) the adjustment procedure for each N/N sensor. The continuous lines identified the results for VD, the dotted ones for the ED SL affected data.

**Table 4.** Statistical parameters for the match-up analysis between the AAOT in situ $R_{rs}$ and the original (adjusted) VD TERRA $R_{rs}$.

| TERRA Bands | $r^2$ | MAD (sr$^{-1}$) | RMSD (sr$^{-1}$) | MAPD (%) |
|---|---|---|---|---|
| $R_{rs}(412)$ | 0.62 (0.84) | $1.3 \times 10^{-3}$ ($6.6 \times 10^{-4}$) | $1.6 \times 10^{-3}$ ($1.0 \times 10^{-3}$) | 31.2 (14.0) |
| $R_{rs}(443)$ | 0.82 (0.89) | $9.1 \times 10^{-4}$ ($6.3 \times 10^{-4}$) | $1.2 \times 10^{-3}$ ($9.0 \times 10^{-4}$) | 19.1 (13.5) |
| $R_{rs}(488)$ | 0.90 (0.90) | $9.2 \times 10^{-4}$ ($8.1 \times 10^{-4}$) | $1.3 \times 10^{-3}$ ($1.2 \times 10^{-3}$) | 13.1 (12.6) |
| $R_{rs}(531)$ | 0.90 (0.90) | $8.5 \times 10^{-4}$ ($7.6 \times 10^{-4}$) | $1.3 \times 10^{-3}$ ($1.2 \times 10^{-3}$) | 11.2 (10.6) |
| $R_{rs}(547)$ | 0.91 (0.92) | $8.4 \times 10^{-4}$ ($7.6 \times 10^{-4}$) | $1.2 \times 10^{-3}$ ($1.2 \times 10^{-3}$) | 11.6 (10.9) |
| $R_{rs}(667)$ | 0.83 (0.87) | $4.1 \times 10^{-4}$ ($2.3 \times 10^{-4}$) | $5.2 \times 10^{-4}$ ($3.5 \times 10^{-4}$) | 38.6 (18.3) |

**Table 5.** Statistical parameters for the match-up analysis between the AAOT in situ $R_{rs}$ and the original (adjusted) VD AQUA $R_{rs}$.

| AQUA Bands | $r^2$ | MAD (sr$^{-1}$) | RMSD (sr$^{-1}$) | MAPD (%) |
|---|---|---|---|---|
| $R_{rs}(412)$ | 0.67 (0.91) | $1.1 \times 10^{-3}$ ($6.1 \times 10^{-4}$) | $1.6 \times 10^{-3}$ ($7.9 \times 10^{-4}$) | 26.5 (13.3) |
| $R_{rs}(443)$ | 0.85 (0.94) | $8.7 \times 10^{-4}$ ($5.7 \times 10^{-4}$) | $1.2 \times 10^{-3}$ ($7.4 \times 10^{-4}$) | 17.8 (12.5) |
| $R_{rs}(488)$ | 0.92 (0.95) | $8.8 \times 10^{-4}$ ($6.4 \times 10^{-4}$) | $1.2 \times 10^{-3}$ ($9.1 \times 10^{-4}$) | 12.1 (9.1) |
| $R_{rs}(531)$ | 0.93 (0.95) | $8.1 \times 10^{-4}$ ($6.6 \times 10^{-4}$) | $1.2 \times 10^{-3}$ ($9.9 \times 10^{-4}$) | 10.6 (8.8) |
| $R_{rs}(547)$ | 0.93 (0.95) | $8.0 \times 10^{-4}$ ($6.9 \times 10^{-4}$) | $1.1 \times 10^{-3}$ ($1.0 \times 10^{-3}$) | 11.0 (9.6) |
| $R_{rs}(667)$ | 0.89 (0.92) | $3.5 \times 10^{-4}$ ($2.1 \times 10^{-4}$) | $4.4 \times 10^{-4}$ ($2.9 \times 10^{-4}$) | 34.5 (18.7) |

**Table 6.** Statistical parameters for the match-up analysis between the AAOT in situ $R_{rs}$ and the original (adjusted) VD VIIRSN $R_{rs}$.

| VIIRSN Bands | $r^2$ | MAD (sr$^{-1}$) | RMSD (sr$^{-1}$) | MAPD (%) |
|---|---|---|---|---|
| $R_{rs}$(410) | 0.51 (0.89) | $1.5 \times 10^{-3}$ ($6.6 \times 10^{-4}$) | $2.1 \times 10^{-3}$ ($8.8 \times 10^{-4}$) | 30.5 (13.3) |
| $R_{rs}$(443) | 0.78 (0.93) | $1.2 \times 10^{-3}$ ($6.0 \times 10^{-4}$) | $1.7 \times 10^{-3}$ ($8.3 \times 10^{-4}$) | 22.9 (12.1) |
| $R_{rs}$(486) | 0.90 (0.94) | $9.2 \times 10^{-4}$ ($6.6 \times 10^{-4}$) | $1.3 \times 10^{-3}$ ($1.0 \times 10^{-3}$) | 12.2 (8.8) |
| $R_{rs}$(551) | 0.94 (0.95) | $7.7 \times 10^{-4}$ ($6.1 \times 10^{-4}$) | $1.1 \times 10^{-3}$ ($9.7 \times 10^{-4}$) | 10.5 (8.6) |
| $R_{rs}$(671) | 0.88 (0.88) | $3.1 \times 10^{-4}$ ($2.6 \times 10^{-4}$) | $4.1 \times 10^{-4}$ ($3.7 \times 10^{-4}$) | 29.8 (24.2) |

**Table 7.** Statistical parameters for the match-up analysis between the AAOT in situ $R_{rs}$ and the original (adjusted) VD VIIRSJ $R_{rs}$.

| VIIRSJ Bands | $r^2$ | MAD (sr$^{-1}$) | RMSD (sr$^{-1}$) | MAPD (%) |
|---|---|---|---|---|
| $R_{rs}$(411) | 0.58 (0.89) | $1.4 \times 10^{-3}$ ($6.0 \times 10^{-4}$) | $1.9 \times 10^{-3}$ ($7.9 \times 10^{-4}$) | 26.6 (11.7) |
| $R_{rs}$(445) | 0.81 (0.92) | $1.0 \times 10^{-3}$ ($6.0 \times 10^{-4}$) | $1.3 \times 10^{-3}$ ($8.2 \times 10^{-4}$) | 18.3 (10.2) |
| $R_{rs}$(489) | 0.90 (0.94) | $9.5 \times 10^{-4}$ ($6.4 \times 10^{-4}$) | $1.3 \times 10^{-3}$ ($9.0 \times 10^{-4}$) | 11.9 (8.3) |
| $R_{rs}$(556) | 0.93 (0.95) | $7.9 \times 10^{-4}$ ($5.6 \times 10^{-4}$) | $1.1 \times 10^{-3}$ ($8.0 \times 10^{-4}$) | 11.3 (8.4) |
| $R_{rs}$(667) | 0.83 (0.88) | $3.3 \times 10^{-4}$ ($1.9 \times 10^{-4}$) | $4.2 \times 10^{-4}$ ($2.7 \times 10^{-4}$) | 34.3 (21.3) |

After the adjustment MAPD is spectrally reduced for each N/N sensor and $r^2$ is spectrally increased except for the 671 nm band for VIIRSN (nearly constant, Figures 9 and 10). Particularly, $r^2$ is now close to or larger than 0.90 for each band of each sensor, with just the exception of the first TERRA band for which $r^2$ is 0.84. For MAPD, a lower decrease is observed in the green part of the spectrum, since the green bands were already in good agreement with the in situ observations before the adjustment. The blue and red bands show the larger improvement: for the first original blue band, MAPDs were larger than 25% for all sensors, while after the adjustment they are lower than 15%; for the red band MAPDs were between 25.0% and 30.0%, and after the adjustment, these decreased to values that go from 18.3% (TERRA) to 24.2% (VIIRSN). Hence, after the adjustment, the agreement between the satellite and in situ spectra is strongly improved, and this is confirmed also by the other statistical parameters (from Tables 4–7) that are spectrally reduced.

The agreement with the in situ spectra differs from one sensor to another. Particularly, TERRA is the one that shows the larger discrepancies with the in situ data both before and after the adjustment procedure. This can be due to the serious degradation of the TERRA mirror; indeed, for this sensor, not even the vicarious calibration has led to the retrieval of reliable OC products, and a calibration based on other functioning OC sensor observations is applied to the TERRA data [3,89]. This degradation can also explain the lower number of valid observations and match-up points for TERRA (218), in comparison with AQUA (357) and VIIRSN (449). Despite that, the adjustment has strongly improved the quality of the TERRA $R_{rs}$ spectra, enabling a more reliable usage of the data from this sensor.

Some differences in the statistics for different sensors can also be partly due to the different spectral resolutions; indeed, in the band shifting procedure the spectral distances between the native in situ (AAOT) and target bands (satellite) are different for different sensors, and larger spectral distances can increase the uncertainty of the band-shifting [88]. This validates the choice to keep the spectral resolution of each VGOCS sensor and to do not band-shift all of them to a set of predefined wavelengths. Indeed, the AAOT central wavelengths varied during the years, and choosing a set of predefined wavelengths for the satellite sensors could lead to the need to band-shift both in situ and satellite data to this set, increasing the uncertainty.

In Figures 11 and 12, the scatterplots for the original and adjusted OLCI $R_{rs}$ in comparison with the in situ data are shown, with the statistical parameters spectra shown in Figure 13 and listed in Table 8. The number of match-up points is only 67 due to the short OLCI time series, the larger revisiting time of Sentinel 3A in comparison with the other VGOCS satellites, and the lack of AAOT data from the end of June to the beginning of October of 2017. The uncertainty observed for $R_{rs}$(412) is larger than those observed for the other OC sensors of the constellation. Despite that, the adjustment largely reduces

this uncertainty, showing similar results to those of the previous analyses (Figure 13). Indeed, after the adjustment $r^2$ is 0.89 for the red band and larger than 0.9 for the other ones, while MAPD is ≈23% for the red band, 10.3% for the 412 nm band, and lower than 10.4% for the other ones.

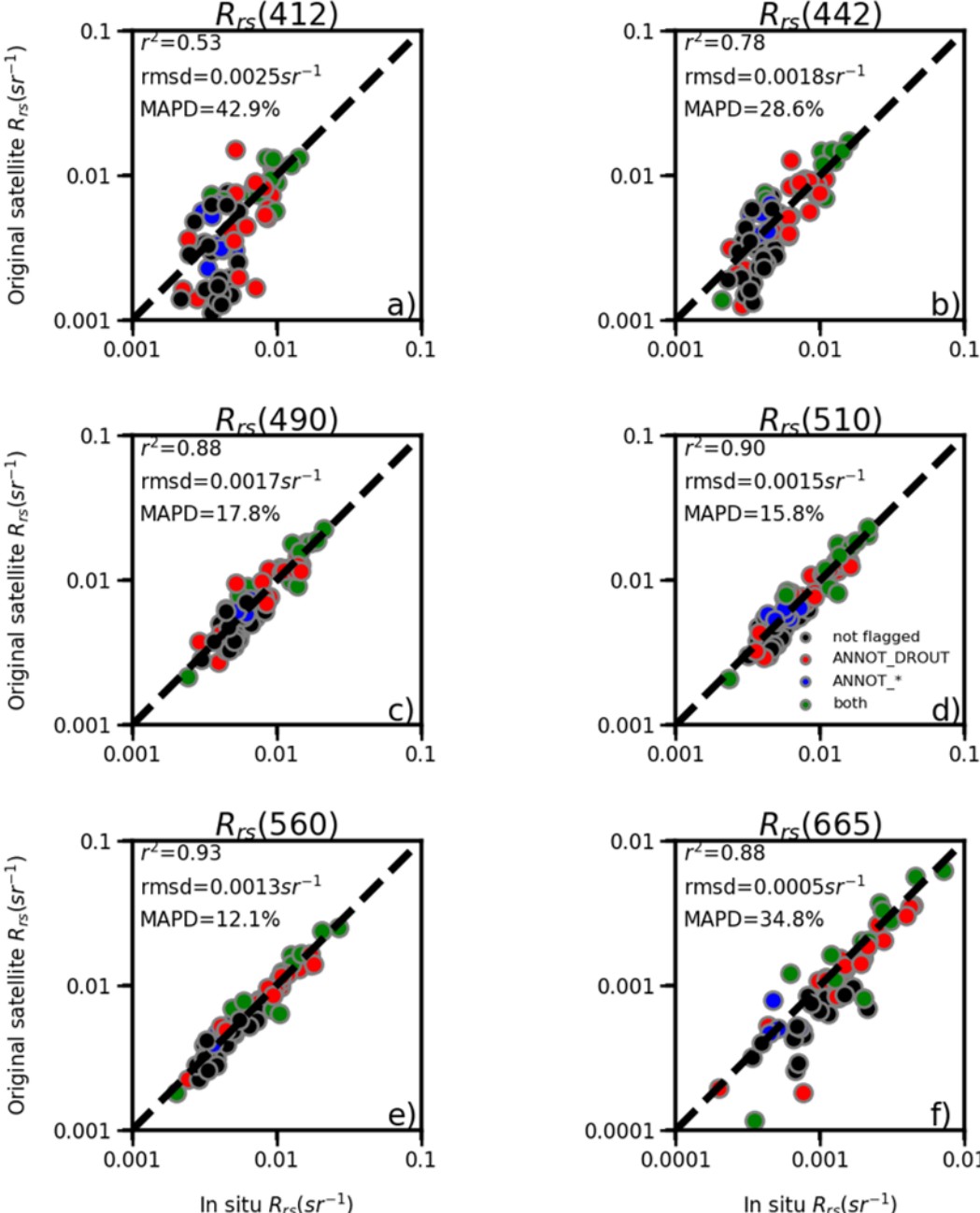

**Figure 11.** Match-up analysis between the in situ $R_{rs}$ and the original VD OLCI $R_{rs}$. In the logarithmic scale on the x-axes the in situ $R_{rs}$ and on the y-axes the original OLCI $R_{rs}$ for each band (**a–f**). The red points are the observations where the ANNOT_DROUT flag is active, the blue points those where the ANNOT_* flags are active, the green ones those where both flags are active, and the black points where none of those flags is active.

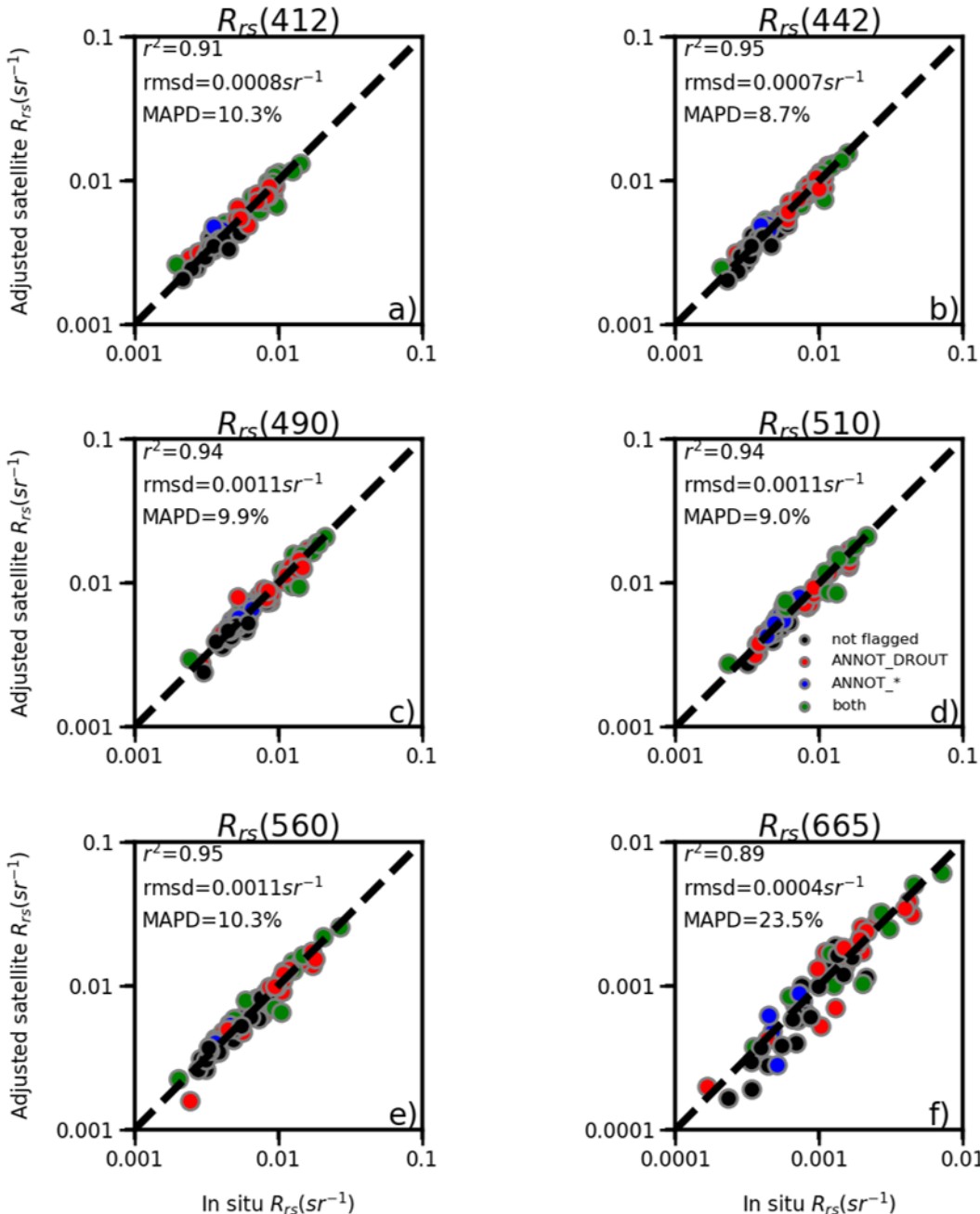

**Figure 12.** Match-up analysis between the in situ $R_{rs}$ and the adjusted VD OLCI $R_{rs}$. In the logarithmic scale on the x-axes the in situ $R_{rs}$ and on the y-axes the adjusted OLCI $R_{rs}$ for each band (**a–f**). The red points are the observations where the ANNOT_DROUT flag is active, the blue points those where the ANNOT_* flags are active, the green ones those where both flags are active, and the black points where none of those flags is active.

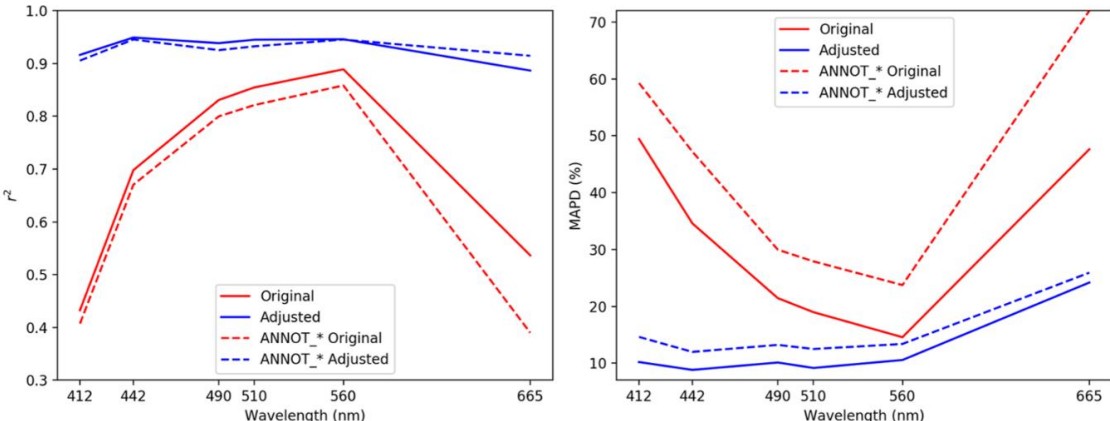

**Figure 13.** Spectra of $r^2$ (left) and MAPD (right) before (red) and after (blue) the adjustment procedure for OLCI. The continuous lines identified the results for VD, the dotted ones for the ANNOT_* data.

**Table 8.** Statistical parameters for the match-up analysis between the AAOT in situ $R_{rs}$ and the original (adjusted) VD OLCI $R_{rs}$.

| OLCI Bands | $r^2$ | MAD (sr$^{-1}$) | RMSD (sr$^{-1}$) | MAPD (%) |
|---|---|---|---|---|
| $R_{rs}(412)$ | 0.43 (0.92) | $2.1 \times 10^{-3}$ ($5.6 \times 10^{-4}$) | $2.8 \times 10^{-3}$ ($7.9 \times 10^{-4}$) | 49.4 (13.3) |
| $R_{rs}(442)$ | 0.70 (0.95) | $1.6 \times 10^{-3}$ ($5.0 \times 10^{-4}$) | $2.2 \times 10^{-3}$ ($7.4 \times 10^{-4}$) | 34.6 (12.5) |
| $R_{rs}(490)$ | 0.83 (0.94) | $1.4 \times 10^{-3}$ ($7.7 \times 10^{-4}$) | $1.9 \times 10^{-3}$ ($9.1 \times 10^{-4}$) | 21.4 (9.1) |
| $R_{rs}(510)$ | 0.85 (0.95) | $1.3 \times 10^{-3}$ ($7.2 \times 10^{-4}$) | $1.8 \times 10^{-3}$ ($9.9 \times 10^{-4}$) | 19.0 (8.8) |
| $R_{rs}(560)$ | 0.89 (0.95) | $1.0 \times 10^{-3}$ ($7.7 \times 10^{-4}$) | $1.6 \times 10^{-3}$ ($1.0 \times 10^{-3}$) | 14.6 (9.6) |
| $R_{rs}(665)$ | 0.54 (0.89) | $4.9 \times 10^{-4}$ ($3.1 \times 10^{-4}$) | $1.0 \times 10^{-3}$ ($2.9 \times 10^{-4}$) | 47.6 (18.7) |

The better agreement between the satellite and in situ spectra strongly increases also the quality of the IOPs retrieved by the QAA [6]. Indeed, the quality of the QAA outputs is dependent on the input $R_{rs}$ spectrum uncertainty; consequently, the reduction of the discrepancies between the satellite and in situ $R_{rs}$ bring to a better agreement between the satellite and in situ QAA parameters [6].

### 3.2. VGCOS Spatial and Temporal Coverage

As it is shown in Section 2.3, the unmasking of the HSZ, SL, ANNOT_DROUT, ANNOT_* data leads to an increase of the VGOCS spatial and temporal coverage. In this section, the agreement of those data with the in situ spectra will be analyzed, to evaluate if they can be unmasked in the VGOCS dataset.

#### 3.2.1. HSZ Flag

The quality of the OC data can be dependent on the viewing geometry and, particularly, on $\theta_v$. The HSZ flag masks data acquired at $\theta_v$ larger than 60° since they can be affected by larger uncertainties in the atmospheric correction and the normalization of the water leaving radiance [69,70]. Nevertheless, also data acquired at large $\theta_v$, but lower than 60° (for example 50° $< \theta v < 60°$) can be affected by larger discrepancies in comparison with in situ or other satellite data [6,38].

To evaluate this $\theta_v$ dependence for the N/N sensors, MAPD has been calculated for different ranges of $\theta_v$ with a step of 10° (Figure 14) for the match-up analyses of the previous section. Before the adjustment, the VIIRSN and VIIRSJ $R_{rs}$ with $\theta_v > 50°$ show larger MAPDs in comparison of those acquired at $\theta_v < 50°$, especially for the 411/2 nm band and the 667/671 ones. After the adjustment MAPD is lower in each bin for each sensor, reducing a general bias with the in situ data, confirming the aggregated results presented in Section 3.1. MAPDs for data acquired at $\theta_v > 50°$ are still slightly larger than those acquired at lower $\theta_v$, but the dependence is now consistently reduced. For example, for the original VIIRSJ 411 nm band MAPD is between 18.1% and 28.5% for the first five bins, while it

is larger than 34% for the last two; after the adjustment, MAPD is between 10.4 and 13% for θv < 50° and it is <14.0% for θv > 50°, confirming the reduced θv dependence.

AQUA and TERRA show similar behavior of the one observed for VIIRSJ/N, but for them also the original data acquired at 40°<θ$_v$<50° show larger discrepancies with the in situ data, but they are strongly reduced with the adjustment. This larger θ$_v$ dependence for AQUA and TERRA can be due to the pixel growth effect [90,91], which can be more evident in the MODIS sensors, as this is reduced in VIIRS by its aggregation algorithm [91]. The main result of this analysis is that now most of the observations acquired at θ$_v$ > 50° have MAPD lower of those acquired at θ$_v$ < 50° before the adjustment.

Consequently, data previously masked by the HSZ flag (θ$_v$ > 60°) can now be exploited, making available a large number of data, usually discarded in standard applications.

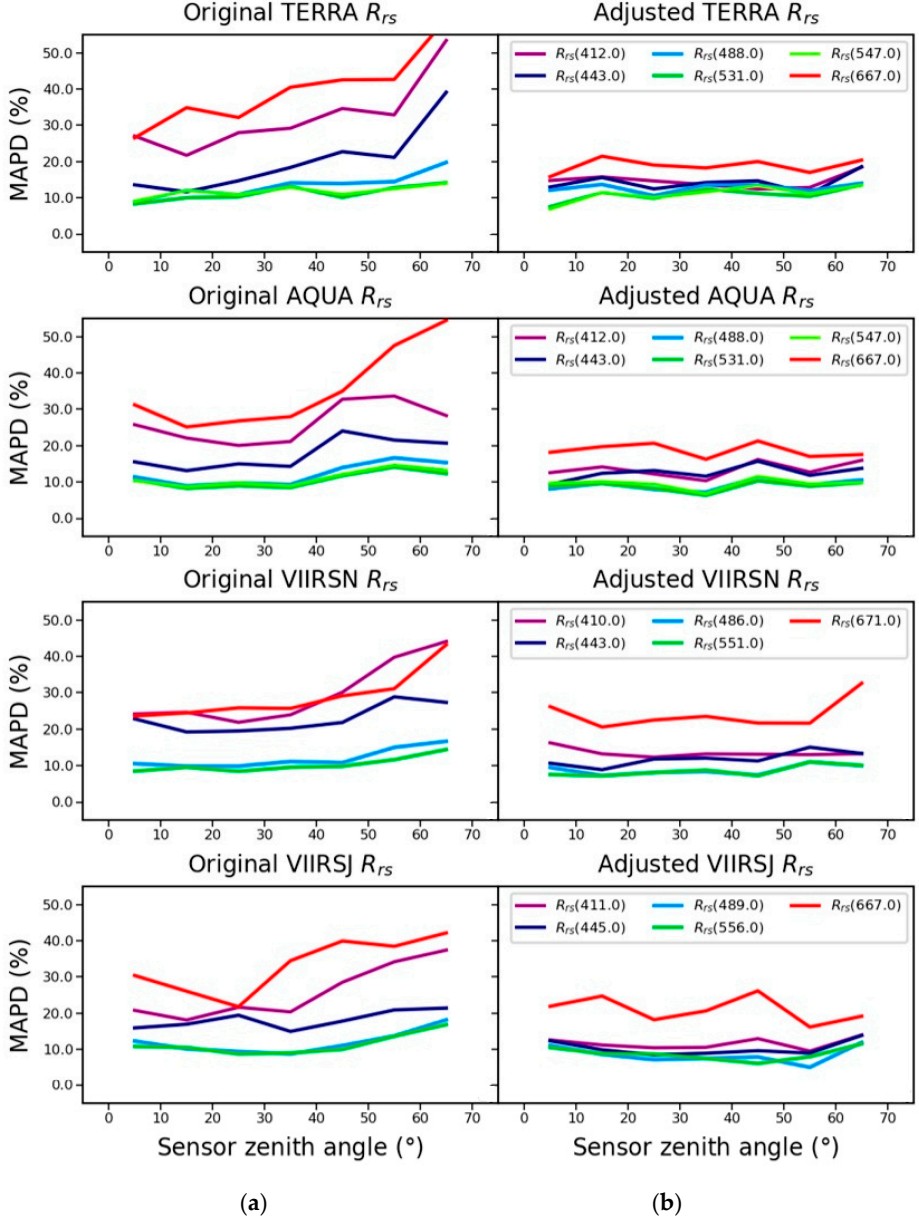

**Figure 14.** Analysis of the MAPD dependence from θ$_v$ before (left, column **a**) and after the adjustment (right, column **b**). On the x-axes the wavelength, on the y-axes MAPD, with the different colours of the lines identifying the different R$_{rs}$ bands.

As already shown in Figures 4 and 5, unmasking for the HSZ flag provides up to 3 additional full images a day of the NAS, strongly increasing the VGOCS spatial and temporal coverage, which is fundamental for the aim for which VGOCS is conceived.

### 3.2.2. SL Flag

The SL affected pixels can have large sources of uncertainties [6,71–74], and, before using them, it is important to test their agreement with the in situ observations. For this reason, match-up analyses for the original and adjusted $R_{rs}$ of the N/N sensors have been performed, considering only the SL affected data (spectra of $r^2$ and MAPD in Figures 9 and 10). For those analyses, for each sensor ED (TD+VD) has been used, to have a larger number of match-up points.

For the original SL affected data (red dotted line), the values of $r^2$ and MAPD are considerably lower and larger respectively than the values observed for VD (red continuous line), confirming that the stray-light effect can significantly lower the quality of the OC data. Nevertheless, after the adjustment $r^2$ and MAPD are considerably closer to 1 and 0 respectively. The behavior of each sensor is similar to the one observed in Section 3.1, with TERRA that still shows larger discrepancies in comparison with the in situ data, due to the degradation of its mirror [3,89]. There are still some residuals due to the stray-light effect, as the MLR algorithm can only minimize and not eliminate the larger uncertainties introduced by this effect. Nevertheless, the values of the statistical parameters (blue dotted lines) are now closer to those observed for VD after the adjustment (blue continuous line). Moreover, the values of MAPD and $r^2$ of the adjusted SL data are respectively lower and larger than those observed for the original $R_{rs}$ in VD.

Hence, as for the HSZ flag (Section 3.2.1), the adjustment makes available such data, allowing to analyze the coastal optical variability and further improving the VGOCS spatial coverage close to shore (Section 2.3.1).

### 3.2.3. ANNOT_DROUT and ANNOT_* Flags

The ANNOT_DROUT and the ANNOT_* flags have not been applied in the OLCI processing chain, as they mask a large number of pixels close to the coast (Section 2.3.2).

As stated earlier, recently EUMETSAT has removed the ANNOT_DROUT flag from the standard L2 flags (I.Cazzaniga, pers.comm.) [83]. The validity of this choice is confirmed by the match-up analyses of Figures 11 and 12, where the comparison between the in situ and original/adjusted OLCI $R_{rs}$ is presented. In these figures, the green and red data are those for which the ANNOT_DROUT flag is active. Not considering them, the number of match-up points is drastically reduced, as only 30 observations are now available in comparison with the 67 of Section 3.1. Masking for ANNOT_DROUT data, MAPD is larger and $r^2$ and RMSD lower (not shown) for all the bands in comparison with the match-up analyses accomplished considering the entire OLCI VD (Section 3.1). The RMSD reduction is not due to a better agreement of this dataset with the in situ data, but simply to the order of magnitude of the masked data. Indeed, only one observation with $R_{rs}(412) > 0.01$ sr$^{-1}$ is not masked by ANNOT_DROUT (blue and black points, Figure 11), leading to a lower value of RMSD, which is an absolute difference, and it is dependent from the order of magnitude of the observations, while MAPD is a relative difference. Hence, the application of the ANNOT_DROUT flag does not allow capturing all the optical variability of the basin. For those reasons, this should not be applied in optically complex waters, as previously stated in [82].

In the OLCI match-up analysis, all the data flagged as ANNOT_* (blue and green points in Figures 11 and 12, with the $r^2$ and MAPD spectra in Figure 13) are due to the ANNOT_MIXR1 flag, as this is the only one which usually results to be applied in coastal waters (Section 2.3.2). The ANNOT_MIXR1 flag data show larger MAPDs (red dotted line) in comparison with those of the entire dataset (red continuous line), but those are strongly reduced after the adjustment (blue dotted line). Still slightly larger discrepancies are shown in comparison with those of the adjusted VD $R_{rs}$

(blue continuous line), but their agreement with in situ data is better than the one of the original $R_{rs}$ of VD (red continuous line), similarly to the SL data.

Hence, the adjusted ANNOT_MIXR1 data should not be masked in the NAS coastal waters, for the OLCI processing baseline v2.23. On the contrary, the ANNOT_TAU6, and ANNOT_ABSO_D data have been finally masked in the VGOCS dataset. Hence, they are not usually applied in coastal waters and the use of such data could result in large uncertainties in open waters, as those data have not been characterized by a match-up analysis.

### 3.3. OLCI Camera Dependence

As stated earlier, the quality of the OLCI $R_{rs}$ spectra can be dependent on the different cameras mounted on the optical bench. Indeed, the presence of different camera implies that different ranges of $\theta_v$ correspond to different detectors; consequently, a plot similar to Figure 14 is presented in Figure 15 but with MAPD calculated for the different camera observations (C1: Camera1, C2: Camera2 ... ).

To distinguish between the different cameras, the detector_index field provided in the OLCI L2 files is exploited. Indeed, this field goes from 0 to 3699 (that are the total number of columns of a single L1 file) and each bin of 740 columns corresponds to a different camera (e.g., from 0 to 739 Camera1, from 740 to 1479 Camera 2 and so on ... ).

The observations from each camera (from C1 to C5) are 17,19,17,8, and 6 respectively. Such numbers of observations are too low to characterize OLCI camera dependence. Nevertheless, using the available match-up points it is possible to observe that the value of MAPDs for the original $R_{rs}$ are consistently different for different cameras. Those differences are strongly reduced after the adjustment, with MAPD showing similar values for different cameras. Hence, also if it is not possible to analyze the OLCI camera dependence, the observed reduction of this effect brought by our adjustment is sufficient for the aim of our study.

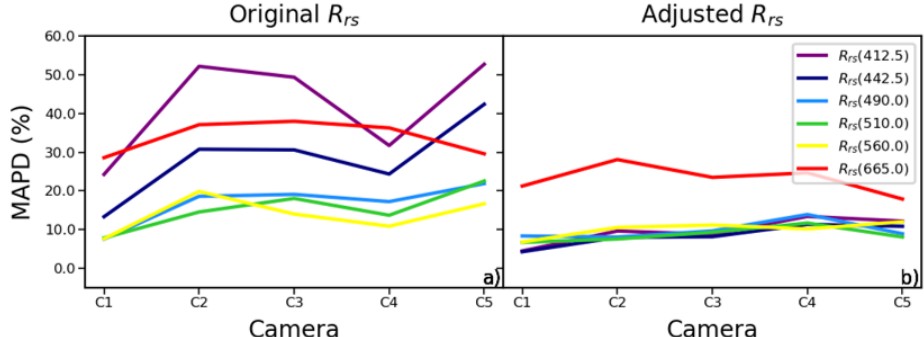

**Figure 15.** Analysis of the dependence of MAPD from the different OLCI cameras for the (**a**) original and (**b**) adjusted $R_{rs}$. On the x-axes the camera (from C1 to C5), on the y-axes MAPD, with the different colours of the lines identifying the different $R_{rs}$ bands.

### 3.4. Inter-Sensor Differences

As described in Section 2.7, to qualify the residual inter-sensor differences, MAPD and $r^2$ have been calculated at the virtual buoys for couples of images acquired in temporal proximity.

In Figures 16 and 17, the spectra of $r^2$ and MAPD at the virtual buoys are presented for the original and adjusted $R_{rs}$, band-shifted to the MODIS bands. $r^2$ is everywhere increased for the two blue bands, while for the other bands it remains almost constant (or slightly reduced), except for the red band at OPEN, where this parameter is reduced. For what concerns MAPD, this is reduced for each band in each location, with a larger decrease observed for the two blue bands and the red band, and a slight increase in the green part of the spectrum at OPEN.

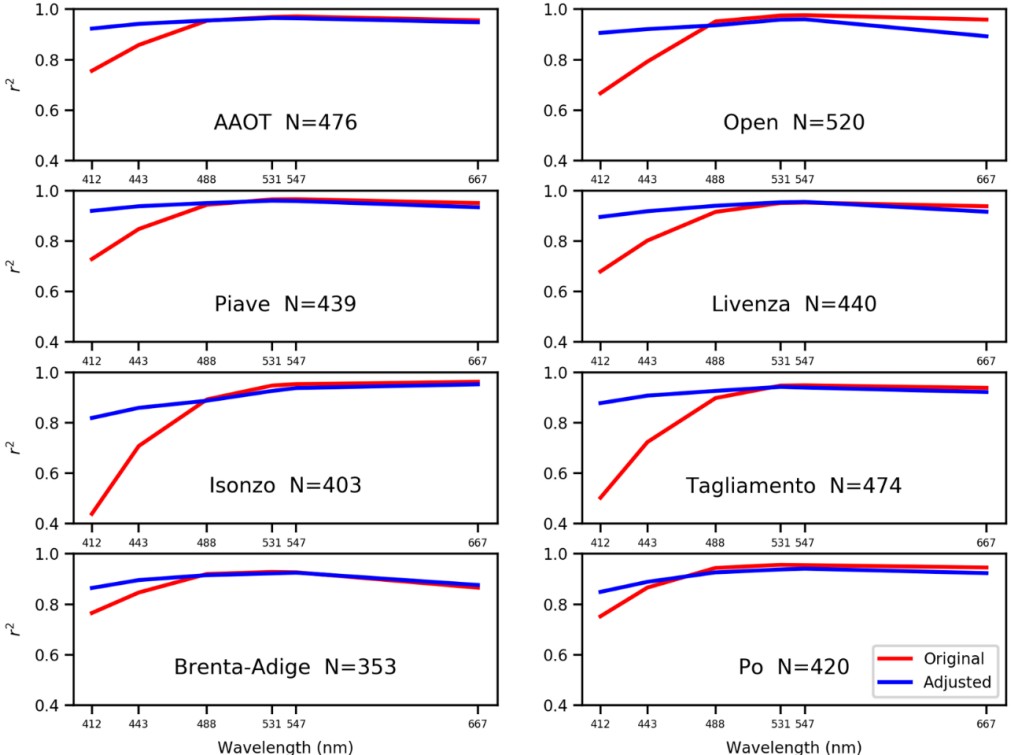

**Figure 16.** The r$^2$ calculated at the virtual buoys for the original (red) and adjusted R$_{rs}$ (blue) for images acquired in temporal proximity by the different VGOCS sensors. Inside the box the virtual buoy name with the number of the couple of images used in the calculation of r$^2$.

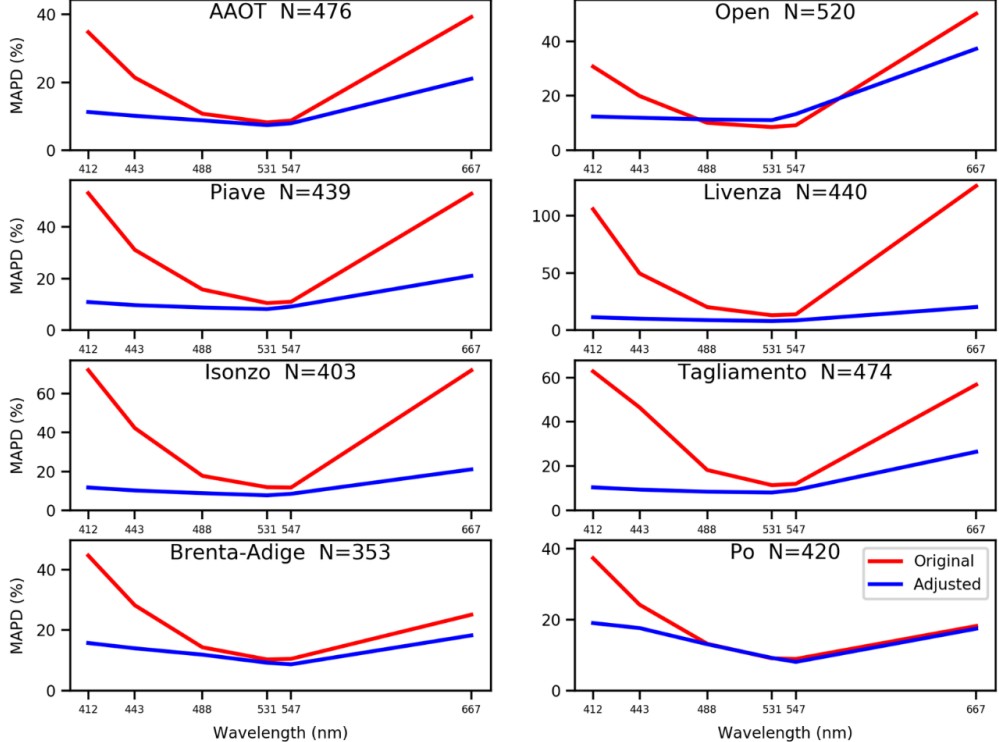

**Figure 17.** Mean absolute percentage difference (MAPD) calculated at the virtual buoys for the original (red) and adjusted R$_{rs}$ (blue) for images acquired in temporal proximity by the different VGOCS sensors. Inside the box the virtual buoy name with the number of the couple of images used in the calculation of MAPD.

The lower reduction/increase of MAPD/ $r^2$ in the 488–547 nm range is due to the good agreement of these bands also for the original $R_{rs}$; thus, the adjustment does not influence too much the $R_{rs}$ retrieval for them, as the original bands are already in good agreement with in situ observations.

The MAPD decrease observed at AAOT in Figure 17 after the adjustment is consistent with the behavior observed for the match-up analyses of Section 3.1, performed used the AAOT in situ data as a reference. For the original sensor match-up analyses, MAPD for the first blue band was between 26.5% and 42.9%, while for the second blue band this was between 17.8% to 28.6%; after the adjustment, those ranges decreased to 10.3–14.0% and 8.7–13.5% respectively. Similarly, for AAOT in the inter-sensor analysis, the values of MAPD decreased from 34.8% and 21.4% to 11.2% and 10.1% for the 412 and 443 nm bands respectively. Moreover, after the adjustment, the MAPD values for the green bands are slightly lower than 10.0%, while the one for the red band is around 20.0%, similarly to the results of Section 3.1.

At the other virtual buoys, the MAPD spectra after the adjustment are similar to the one observed for AAOT, while for the original $R_{rs}$ very different spectra were observed for each of them. Indeed, for the original $R_{rs}$, the ranges of MAPD for the blue bands were between 30.7% (OPEN) and 105.8% (Livenza) and between 19.9% (OPEN) and 49.6% (Livenza). After the adjustment, for the first band, MAPD is between 10.4% (Tagliamento) and 19.0% (Po), while for the second one it is between 9.4% (Tagliamento) and 17.4% (Po). Not considering Po and Brenta-Adige, the superior limits of the two ranges reduce to 12.4% and 11.9% (OPEN), which are values similar to those observed for AAOT in both match-up and inter-sensor analysis. Similarly, for the green bands, all values of MAPD are around 10.0%, while for the red band MAPD is always around 20.0%, except for OPEN, where an MAPD of 37.3% is observed.

Different values of MAPD are observed for the blue bands at Po/Brenta-Adige and the red band at OPEN in comparison with the other virtual buoys since such locations represent the extreme situation for optically complex and open waters respectively, and consequently, the adjustment can be less effective for such type of waters at certain wavelengths.

Similar values of the statistical parameters for the match-up and inter-sensor analysis imply that the residual differences now observed between different satellite images are mostly due to the uncertainty in the atmospheric correction and the absolute calibration of the space sensors; consequently, those that are due to the different technical characteristics, resolutions, viewing geometry, and processing chains are notably reduced. Moreover, these findings confirm the underlying assumption of this study, i.e., that the AAOT in situ data can be exploited to adjust the $R_{rs}$ spectra of the entire basin [6,87].

## 4. Discussion

This study tested the effect of an adjustment on the quality of the satellite data, the VGOCS spatial and temporal coverage, and the inter-sensor differences to demonstrate the suitability of VGOCS in analyzing the coastal optical variability.

Several match-up analyses between the original $R_{rs}$ of different satellite sensors and the AAOT $R_{rs}$ data were performed. The adjustment brought to a significant improvement of the satellite $R_{rs}$ retrieval with a large spectral increase of $r^2$ and a spectral reduction of the other statistical parameters for all the VGOCS sensors. The better agreement between the satellite and in situ spectra strongly increases also the quality of the IOPs retrieved by the QAA, which are needed to analyze the coastal optical variability [6].

The adjustment was exploited also to make available data generally masked in the standard processing chains. Indeed, data acquired at $\theta v > 60°$ are usually masked by the application of the HSZ flag, due to larger uncertainties [38]. This is confirmed by the analysis of the dependence of the quality of the original $R_{rs}$ spectra on $\theta_v$; indeed, for the original $R_{rs}$ larger discrepancies were observed for high $\theta_v$, confirming the results of previous works [6–38]. Nevertheless, the adjustment reduced the biases between the satellite and in situ data for all ranges of $\theta_v$, especially for those acquired at $\theta_v > 50°$,

leading to a strong reduction of the $\theta_v$ dependence. This enabled to use also data usually masked by the HSZ flag, improving the VGOCS temporal and spatial coverage.

The VGOCS spatial coverage is further increased by the use of data usually masked by the SL flag for the N/N sensors and by the ANNOT_DROUT and ANNOT_MIXR1 flags for OLCI.

The adjustment strongly reduces the uncertainty for the SL affected pixels and it allows to have more valid pixels close to the shore, as the application of this flag leads to the masking of the 40% of the coastal pixels. In the future, to additionally reduce the SL data uncertainty, the calculation of a different set of adjustment coefficient could be considered, using in the TD only SL data.

The analysis of the ANNOT_DROUT data confirms the findings of [82] and the validity of the choice by EUMETSAT to remove this flag from the list of the standard L2 flags (I. Cazzaniga, pers.comm.) [83], as they result to be in good agreement with in situ data in the coastal area and they allow to analyze the coastal optical variability.

The behavior of the ANNOT_MIXR1 data, which are 20.1% of the OLCI coastal pixels, is similar to the SL ones, thus they have been unmasked. On the contrary, as it is not possible to characterize the ANNOT_TAU6 and ANNOT_ABSO_D data, they have been masked in our dataset, as they are not encountered in the coastal area and they can show large uncertainties in open waters.

The VGOCS spatial and temporal coverage will be further increased in the future. Hence, in the dataset, only the sensors that are currently functioning were included, as the present days are those with a larger number of sensors on orbit. Nevertheless, in the future also other sensors, that are not anymore functioning, such as the Sea-Viewing Wide Field-of-View Sensor (SeaWiFS) mounted on the satellite SeaStar and MERIS mounted on the Envisat satellite, will be included. Moreover, when the vicarious calibration process will be accomplished, also the OLCI Sentinel 3B sensor will be added to the constellation.

The analysis of the $\theta_v$ dependence was not performed for OLCI as different ranges of $\theta_v$ are scanned by different cameras. The low number of match-up points does not allow to deeply analyze the camera dependence, but different values of MAPD were observed for different cameras. After the adjustment, the difference between different cameras is strongly reduced and MAPD is similar for each of them. In the future, to deeply characterize this effect, the analysis can be performed simply enlarging the $\Delta t$ of the match-up. Indeed, in this study $\Delta t$ has been chosen as very narrow (30 min) to take into account the large temporal variability of the coastal area.

Starting from the hypothesis that no considerable oceanographic processes can occur in 10 min, the differences between $R_{rs}$ spectra measured by different sensors have been estimated using images acquired in temporal proximity. The values of MAPD observed in this analysis after the adjustment, similar to those observed in the comparison with the in situ data, proved that the MLR scheme notably reduced the inter-sensor differences, which can lead to a misleading interpretation of the basin optical variability when two images from different sensors are compared.

In the introduction, the VGOCS has been defined as a hyper-temporal dataset, but to be coherent with this definition, the dataset must fulfill the three conditions defined in [29] and reported in Section 1:

(a)   The variables stored in the dataset must be univariate. As the $R_{rs}$ data are provided at their native spectral resolution to reduce the uncertainty in the band-shifting procedure, the variables stored in the $R_{rs}$_data group are different for each sensor and this condition seems to be not fulfilled. Nevertheless, the $R_{rs}$ data group is mainly provided for post-processing procedures and it allows the users to calculate different IOPs and water component concentrations using different algorithms [92–95]. The parameters that are needed to analyze the coastal optical variability, the aim for which VGOCS is conceived, are the IOPs, that are the same for each sensor, fulfilling the first requirement.

(b)   All the images must be time referenced and the same grid should be used for each image of the dataset. This is fulfilled because the name of each file gives information about the acquisition time, and each image of the dataset is reprojected on a standard 1 km × 1 km equirectangular grid.

(c)    All the sensors must be inter-validated between each other. This condition is also reached, as they are all adjusted on the AERONET-OC AAOT in situ data and the adjustment significantly reduced the inter-sensor differences.

## 5. Conclusions

This work aims to fill the gap between the near-polar orbiting and geostationary sensor temporal resolutions over the European coastal basins by the creation of a hyper-temporal Ocean Color analysis-ready dataset, named Virtual Geostationary Ocean Color Sensor (VGOCS). The latter fulfills the conditions needed to be defined as a hyper-temporal dataset and contains observations acquired from several near polar-orbiting Ocean Color imagers over the North Adriatic Sea, allowing approaching the geostationary sensor temporal resolution.

The problem in using data from different sensors is that they are characterized by different uncertainty sources that can introduce artifacts between different satellite images. For this reason, a multi-linear regression adjustment, based on a Fiducial Reference Measurements data-stream, has been applied to the remote sensing reflectance spectra of the VGOCS sensors. Hence, this adjustment, besides improving the agreement between the satellite and in situ data, reduced the inter-sensor differences, allowing the use of data from different sensors in analyzing the coastal optical variability. Moreover, the adjustment made available data traditionally masked in the standard processing chains, strongly increasing the VGOCS spatial and temporal coverage in the coastal area, an improvement that is fundamental to properly capture most of the coastal optical variability.

The approach here presented can be extended to all the coastal areas where one or more automatic radiometric stations, that represent the entire basin optical properties, are present. In European waters, this approach could be extended to:

- The Western Black Sea where the Gloria, Section-7, and Galata Platform AERONET-OC stations are present [96].
- In the Northern Sea where the Zeebrugge-MOW1 and Thornton_C-Power, part of the AERONET-OC network, are located [97].
- In the Baltic Sea where the Helsinki Lighthouse, Irbe Lighthouse, and Gustav Dalen Tower AERONET-OC stations are present [19]. As this basin is located at higher latitudes than the NAS, the satellite overpasses above such area are more frequent, making the approach also more powerful. Nevertheless, due to a large amount of CDOM present in these waters [98], a different atmospheric correction is needed to retrieve reliable Rrs spectra [99–101].
- The North-Western Mediterranean where the BOUee pour l'aquiSition d'une Serie Optique a Long termE mooring (BOUSSOLE) [102] and the Casablanca AERONET-OC [103] station are located. Particularly, BOUSSOLE has been used for the system vicarious calibration of the OLCI sensor [104]; hence, it could be interesting to test the effect of the adjustment using such data as input of the MLR algorithm.
- The areas where the different WATERHYPERNET network sites are located [105]. This network, which will be developed in the next years, is based on the concept of AERONET-OC [26], but with the essential advantage of the exploitation of a hyperspectral radiometer [106]. The use of such an instrument will allow validating each band of each satellite mission [106], reducing the uncertainties that could be introduced by band-shifting procedures [88,107].

The use of the Fiducial Reference Measurements data stream for the creation of this dataset for different coastal basins can be a reliable and unique way to compensate for the lack of an OCR geostationary sensor. Hence, the VGOCS dataset, available upon request to the authors, allows non-expert users to have at the disposal "ready to use" OC data to analyze the coastal optical variability at high temporal resolution. Moreover, expert users can perform analysis of exploiting products usually not provided in standard L3 files, such as flags and viewing geometry parameters.

**Author Contributions:** The general conception of this work was developed by M.B. and V.E.B.; M.B., G.V., M.B.E, and S.C. designed and implemented the processing chain. M.B. and V.E.B. designed the data analysis. M.B. processed the satellite imagery and performed the data analysis. M.B. and V.E.B. wrote the manuscript. S.C., G.V., and R.S. contributed to the interpretation of the results. All co-authors provided critical comments to the manuscript. All authors have read and agreed to the published version of the manuscript.

**Funding:** This research was supported by the European Union's Copernicus Marine Environment and Monitoring Service (grant number: 77-CMEMS-TAC-OC) and the European Union's Horizon 2020 research and innovation programme (HYPERNETS project, Grant agreement n° 775983).

**Acknowledgments:** The AERONET Team is acknowledged for the continuous effort in supporting the AERONET-OC sub-network. Giuseppe Zibordi from the Joint Research Center of the European Commission is acknowledged for establishing and maintaining the AAOT AERONET-OC site. Vega Forneris and Flavio Lapadula from ISMAR-CNR are acknowledged for advice on the implementation of the processing chain. Ilaria Cazzaniga provided relevant information enabling the OLCI flag and camera dependence analysis. Francesco Bignami, Davide D'Alimonte, Davide Dionisi, Federico Falcini, Michela Sammartino, Enrico Zambianchi, and Giuseppe Zibordi, provided useful comments on earlier versions of this manuscript.

**Conflicts of Interest:** The authors declare no conflict of interest.

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
