# Peer review of "A Virtual Geostationary Ocean Color Sensor to Analyze the Coastal Optical Variability"

_remotesensing, doi:10.3390/rs12101539_

Round 1

Reviewer 1 Report

In the manuscript titled “A Virtual Geostationary Ocean Colour Sensor to analyse the coastal optical variability”,  a synthesized ocean color dataset are produced from several ocean color sensors, and in-situ measurements over North Adriatic Sea. Due to different sensor characteristics, processing procedures, scarcity of the collocated datasets, the task is not easy. The authors provided a thorough discussion of the workflow they employed and the various issues and solutions they faced and solved. The manuscript is really valuable for the ocean color community to embrace the full spectrum of the ocean color remote sensing datasets provided by different agencies. The manuscript is well written. I found it a good reading when I have a question and then the authors always provide some discussions. I have only minor comments and suggestions for the authors to consider.

The authors provide a Rrs adjustment procedure for each sensor using regressions, and consider explicitly the dependency on theta_v, thata_s and phi. Therefore the method can corrected the residual dependency on the geometries after BRDF correction. The authors are able to utilize the pixels at high sensor zenith (HSZ) with an angle larger than 60 degrees, and show improved agreement with in-situ measurements after adjustment.  Figure 9 and 10 shows better agreement between sensor and in-situ, this approve that the adjustments are correct. It is also nice to see correction improve inter-sensor consistency for the pixels not collocated with the AERONET site (3.4).

Page 11, line 300 The authors pointed out BRDF correction is important for this study, where NASA OC processing flow have conducted BRDF while it is not built in for OLCI data and they have to consider in their own workflow. But since each sensor will be adjusted anyway in their workflow, is the step for BRDF correction still necessary?

Page 13, Eq (2), why the correction on one wavelength lambda depend on a summation with respect to multiple wavelength lambda_i? Would the correction for each wavelength independent to each other?

Page 4, line 126, it would be useful to provide some information of the water depth here.

Page 5, line 150: what is the salt and pepper effect?

Author Response

In the manuscript titled “A Virtual Geostationary Ocean Colour Sensor to analyse the coastal optical variability”,  a synthesized ocean color dataset are produced from several ocean color sensors, and in-situ measurements over North Adriatic Sea. Due to different sensor characteristics, processing procedures, scarcity of the collocated datasets, the task is not easy. The authors provided a thorough discussion of the workflow they employed and the various issues and solutions they faced and solved. The manuscript is really valuable for the ocean color community to embrace the full spectrum of the ocean color remote sensing datasets provided by different agencies. The manuscript is well written. I found it a good reading when I have a question and then the authors always provide some discussions. I have only minor comments and suggestions for the authors to consider.

The authors provide a Rrs adjustment procedure for each sensor using regressions, and consider explicitly the dependency on theta_v, thata_s and phi. Therefore the method can corrected the residual dependency on the geometries after BRDF correction. The authors are able to utilize the pixels at high sensor zenith (HSZ) with an angle larger than 60 degrees, and show improved agreement with in-situ measurements after adjustment.  Figure 9 and 10 shows better agreement between sensor and in-situ, this approve that the adjustments are correct. It is also nice to see correction improve inter-sensor consistency for the pixels not collocated with the AERONET site (3.4).

We’d like to thank the reviewer for the positive comments about our work and the insightful suggestions.

Page 11, line 300 The authors pointed out BRDF correction is important for this study, where NASA OC processing flow have conducted BRDF while it is not built in for OLCI data and they have to consider in their own workflow. But since each sensor will be adjusted anyway in their workflow, is the step for BRDF correction still necessary?

This is a good consideration and the adjustment can surely compensate for the lack of BRDF correction. Nevertheless, when it was possible we always tried to be coherent between different processing chains to do not introduce additional artefacts. Indeed, we’d like to have at disposal original Rrs data as good as possible, to do not overestimate the effect of the adjustment.

Page 13, Eq (2), why the correction on one wavelength lambda depend on a summation with respect to multiple wavelength lambda_i? Would the correction for each wavelength independent to each other?

In the choice of the input variables, we followed the work of D’Alimonte et al. (2007). Here, it has been demonstrated that the set of input variables more suitable for the adjustment is given by the Lwn entire spectrum (Rrs in our study) and viewing geometry parameters.

Page 4, line 126, it would be useful to provide some information of the water depth here.

This information has been added to the sentence (now line 134-135).

Page 5, line 150: what is the salt and pepper effect?

An explanation of this effect has been provided (lines 159).

Reviewer 2 Report

The manuscript deals with relevant strategies proposed to make more readily available coastal satellite information, in particular ocean color, by means of overlaying and correcting Ocean Colour Radiometry Polar Sensors data. The manuscript is well written and structured, easy to follow and the contribution to the current state of knowledge stands out well from the discussion and conclusions.
Except for minor typos or a couple of badly constructed phrases, as well as the persistent problem with the referencing, the manuscript is recommended for publication as is.

Author Response

The manuscript deals with relevant strategies proposed to make more readily available coastal satellite information, in particular ocean color, by means of overlaying and correcting Ocean Colour Radiometry Polar Sensors data. The manuscript is well written and structured, easy to follow and the contribution to the current state of knowledge stands out well from the discussion and conclusions.
Except for minor typos or a couple of badly constructed phrases, as well as the persistent problem with the referencing, the manuscript is recommended for publication as is.

We wish to thank the reviewer for the positive comments about our work. Some typos have been corrected and the issue with the cross-reference has been fixed.

Reviewer 3 Report

Dear Editor and Authors, the paper is well structured and well written. A small critic regards the length of the paper, which is quite long: some details and informations may have been avoided to underlyne the most important things. Nevertheless, this is subjective, I leave to the Authors the suggestion of shortening the paper.

Unfortunately, the cross-ference didn't work and it was difficult to follow the last part of Results, where Figures were wrong numbered in the text. 

My requirements regard Abstract and Conclusions. Please, consider to rewrite the Abstract introducing the general topic (1-2 sentences), the description/motivation of the paper (3-4 sentences), Methods(2-3 sentences), Results and Conclusions(4-5 sentences).

Conclusions should be concise, the reader must be able to understand your work and findings. Currently, the section does not focus on your important fundings and achievements, and many parts should be moved to Disucssion. 727 742 cite conditions which you should remind to the reader. Conclusions section rules:it must not have references and acronyms must be re-defined. 

Figures are of good quality, apart Fig. 3, which is uselessly huge, it can be plotted better (smaller font) and with some more details added besides the boxes 

Author Response

Dear Editor and Authors, the paper is well structured and well written. A small critic regards the length of the paper, which is quite long: some details and informations may have been avoided to underlyne the most important things. Nevertheless, this is subjective, I leave to the Authors the suggestion of shortening the paper.

We wish to thank the reviewer for the positive comments about our work and the insightful suggestions.

Unfortunately, the cross-ference didn't work and it was difficult to follow the last part of Results, where Figures were wrong numbered in the text. 

The issue with the cross-reference has been fixed.

My requirements regard Abstract and Conclusions. Please, consider to rewrite the Abstract introducing the general topic (1-2 sentences), the description/motivation of the paper (3-4 sentences), Methods(2-3 sentences), Results and Conclusions(4-5 sentences).

The abstract has been re-organized and re-written, following your suggestions.

Conclusions should be concise, the reader must be able to understand your work and findings. Currently, the section does not focus on your important fundings and achievements, and many parts should be moved to Disucssion. 727 742 cite conditions which you should remind to the reader. Conclusions section rules:it must not have references and acronyms must be re-defined. 

The conclusions are now briefer and contains only the most important fundings of the study. The discussion about the fulfilment of the three conditions has been moved (lines 730-744 on the previous version, now lines 720-738). The three conditions have now been made more explicit. Also, the discussion about the sensors that will be included in VGOCS in the future has been moved to Section 4 (previously lines 745-752, now lines 700-706). Acronyms have been re-defined. We did not find an indication about the use of references in the conclusion in the instructions for authors. Nevertheless, we kept only references for the locations and data stream that can be used for the creation of VGOCS in other basins.

Figures are of good quality, apart Fig. 3, which is uselessly huge, it can be plotted better (smaller font) and with some more details added besides the boxes. 

We added more details besides the boxes and reduced the size of the figure.